# A single cell characterisation of human embryogenesis identifies pluripotency transitions and putative anterior hypoblast centre

Matteo A. Molè [1,10,12], Tim H. H. Coorens [2,12], Marta N. Shahbazi [1,11,12], Antonia Weberling [1,12], Bailey A. T. Weatherbee [1,12], Carlos W. Gantner [1], Carmen Sancho-Serra[2], Lucy Richardson[3], Abbie Drinkwater[3], Najma Syed[3], Stephanie Engley[3], Philip Snell[4], Leila Christie[4], Kay Elder [4], Alison Campbell[5], Simon Fishel[5,6], Sam Behjati [2,7,8✉], Roser Vento-Tormo [2✉] & Magdalena Zernicka-Goetz[1,9✉]

Following implantation, the human embryo undergoes major morphogenetic transformations that establish the future body plan. While the molecular events underpinning this process are established in mice, they remain unknown in humans. Here we characterise key events of human embryo morphogenesis, in the period between implantation and gastrulation, using single-cell analyses and functional studies. First, the embryonic epiblast cells transition through different pluripotent states and act as a source of FGF signals that ensure proliferation of both embryonic and extra-embryonic tissues. In a subset of embryos, we identify a group of asymmetrically positioned extra-embryonic hypoblast cells expressing inhibitors of BMP, NODAL and WNT signalling pathways. We suggest that this group of cells can act as the anterior singalling centre to pattern the epiblast. These results provide insights into pluripotency state transitions, the role of FGF signalling and the specification of anterior-posterior axis during human embryo development.

[1] Department of Physiology, Development and Neuroscience, Mammalian Embryo and Stem Cell Group, University of Cambridge, Cambridge, UK. [2] Wellcome Sanger Institute, Hinxton, UK. [3] Herts & Essex Fertility Centre, Bishops College, Cheshunt, Herts, UK. [4] Bourn Hall, Cambridge, UK. [5] CARE Fertility Group, Nottingham, UK. [6] School of Pharmacy and Biomolecular Sciences, Liverpool John Moores University, Liverpool, UK. [7] Cambridge University Hospital, NHS Foundation Trust, Cambridge, UK. [8] Department of Paediatrics, University of Cambridge, Cambridge, UK. [9] Division of Biology and Biological Engineering, California Institute of Technology, Pasadena, CA, USA. [10] Present address: Babraham Institute, Babraham Research Campus, Cambridge, UK. [11] Present address: MRC Laboratory of Molecular Biology, Cambridge, UK. [12] These authors contributed equally: Matteo A. Molè, Tim H. H. Coorens, Marta N. Shahbazi, Antonia Weberling, Bailey A. T. Weatherbee. ✉email: sb31@sanger.ac.uk; rv4@sanger.ac.uk; mz205@cam.ac.uk

The development of the human embryo during the second week of gestation represents a critical stage of embryogenesis as the free-floating blastocyst implants into the maternal endometrium while undergoing major morphogenetic transformations in preparation for the subsequent phase of gastrulation (14 days post-fertilisation (d.p.f.)). Key events include the transition in the pluripotency state and generation of the post-implantation embryonic epiblast epithelium, followed by a symmetry-breaking event that leads to the formation of the anterior–posterior (AP) body axis[1–5]. Concomitantly, the extra-embryonic tissues undergo a global reorganisation, with the hypoblast making the primary yolk sac and the trophoblast differentiating into the major cell types of the placenta. These morphogenetic transformations need to be tightly coordinated. Failure of development during this time represents one of the major causes of early pregnancy loss in patients undergoing assisted conception treatments[6]. Despite this relevance, we currently lack a functional understanding of the molecular and cellular mechanisms governing this stage of human development.

Although embryo development is not amenable to in vivo research in humans, the use of model systems such as mouse and monkey has led to the identification of the molecular and cellular mechanisms that drive symmetry breaking and specification of the AP axis[7–13]. In mouse embryos, a subset of distal extra-embryonic endoderm cells is specified as the distal visceral endoderm (DVE), which migrates towards the future anterior domain giving rise to the anterior visceral endoderm (AVE)[14–19]. These AVE cells secrete NODAL, BMP and WNT inhibitors, leading to a gradient of signalling activity across the epiblast that defines the AP axis[13]. Interestingly, in cynomolgus monkey embryos, all hypoblast cells initially express the BMP inhibitor CER1, which progressively becomes restricted to the anterior side of the embryo[11]. These findings indicate that the mechanisms leading to symmetry breaking are not completely conserved across species, and raise the question of how epiblast patterning is initiated in human embryos.

The recent establishment of methods to culture human embryos beyond implantation in vitro offers an unprecedented opportunity to characterise the molecular and cellular transformations that human embryos undergo during the second week of development[20–22]. These methodologies have been used to describe the transcriptional profiles of embryos at different stages of development[23–25]. However, functional and in-depth characterisation experiments to determine how signalling interactions between embryonic and extra-embryonic cells reshape the developing human embryo are lacking.

Here, we combine single-cell sequencing and functional experiments to investigate the signalling interactions between embryonic and extra-embryonic cells that underlie the development of the in vitro cultured human embryo between implantation and gastrulation. We observe that the epiblast transitions through different pluripotent states and acts as a source of FGF signals necessary for proliferation of all the lineages during early post-implantation development. At 9 d.p.f., a subset of hypoblast cells reminiscent of the anterior visceral endoderm in the pre-gastrulation mouse embryo emerges. These cells secrete conserved inhibitors of the NODAL, BMP and WNT signalling pathways, and could initiate patterning of the anterior-posterior axis prior to gastrulation.

## Results

### Single-cell RNA sequencing of human embryos at 9 and 11 d.p.f.
To determine the spatio-temporal gene expression profile of the human embryo beyond implantation, we cultured blastocysts, donated from patients undergoing in vitro fertilisation (IVF)

treatment, from day 5 d.p.f. until 9 and 11 d.p.f. . We analysed a total of 29 embryos by single-cell RNA sequencing (scRNA-seq) using 10x Genomics Chromium[26] (Supplementary Data 1). 13 out of 29 embryos lacked inner cell mass (ICM) derivatives, either embryonic epiblast cells or extra-embryonic hypoblast cells, and were therefore excluded from downstream analyses. Our final dataset comprised 16 embryos (8 at 9 d.p.f. and 8 at 11 d.p.f.) and a total of 4820 cells (Supplementary Fig. 1a–g, Supplementary Data 1).

We defined four distinct clusters based on the expression of known markers (Fig. 1a): the epiblast, precursor of the embryo proper (cluster 1, 166 cells), the hypoblast, precursor of the yolk sac (cluster 2, 136 cells), and the trophoblast, which mediates implantation of the embryo into the maternal endometrium and contributes to the formation of the placenta (clusters 3–4). Based on differential expression gene analysis, we highlighted the most significantly enriched genes defining each lineage (Fig. 1b, bold red, Supplementary Data 2, Supplementary Fig. 2a–d). The epiblast was characterised by the expression of the canonical pluripotency genes POU5F1, NANOG and SOX2, as well as other markers such as DPPA5, KHDC1L, MT1X, KHDC3L and MT1G. The hypoblast could be distinguished by the expression of the canonical markers PDGFRA and GATA6, alongside other genes such as APOA1, FN1, S100A14, COL4A1 and APOA2. The trophoblast appeared to develop into two subpopulations: the cytotrophoblast lineage (cluster 3, 2182 cells), identified by the expression of PEG10, FABP5, HIST1H4C, KRT19, TPM1, the E-CADHERIN gene CDH1 and the integrin subunit ITGA6; and the syncytiotrophoblast (cluster 4, 2336 cells), marked by the expression of the markers CGA, PRR9, ANXA1, LGALS16 and ATF3, similar to expression profiles reported elsewhere[27–31]. Interestingly, the mouse trophoblast marker EOMES was not detected in agreement with previous findings in human[23,24] (Supplementary Fig. 2 e, f) and monkey embryos[8] (Supplementary Fig. 2 g, h). Next, we projected previously published single cell expression profiles of human embryos cultured in vitro at equivalent stages using a logistic regression model trained on our embryo data[23,24]. This analysis revealed a high cluster similarity between the datasets (Supplementary Fig. 3a, b) and confirmed the identification of the major cell types present in post-implantation human embryos. We provide an open-source web server at www.humanembryo.org (see 'Methods').

**Pluripotency transition from pre to post-implantation.** Exit from the unrestricted naïve pluripotent state and transition to the lineage-biased primed pluripotent state occurs in mouse and monkey epiblasts as the embryo develops from pre- to post-implantation[7,8]. This transition is required for correct morphogenesis of the epiblast lineage following implantation[32]. To characterise the transcriptional changes associated with pluripotency transition in the human embryo developing in culture, we integrated our post-implantation epiblast datasets at 9 and 11 d.p.f. with recently generated single-cell datasets corresponding to the pre-implantation ICM at 5 d.p.f. and epiblast at 6-7 d.p.f.[33–36] (Fig. 1c, Supplementary Data 4). We found that major naïve markers KLF4, KLF17, PRDM14, DNMT3L, SOX15, TFCP2L1, ZFP42 were all highly expressed in the pre-implantation epiblast and became downregulated post-implantation at 9 and 11 d.p.f. (Fig. 1d). An exception was the gene UTF1, which became upregulated instead. Conversely, major primed pluripotency markers FGF2, DNMT3B, SOX11, SFRP2 and SALL2 showed an increased expression in the post-implantation epiblast (Fig. 1e). The mouse early post-implantation epiblast markers FGF5 and OTX2[37,38] exhibited low expression levels in ICM and epiblast cells throughout the different developmental states. Interestingly,

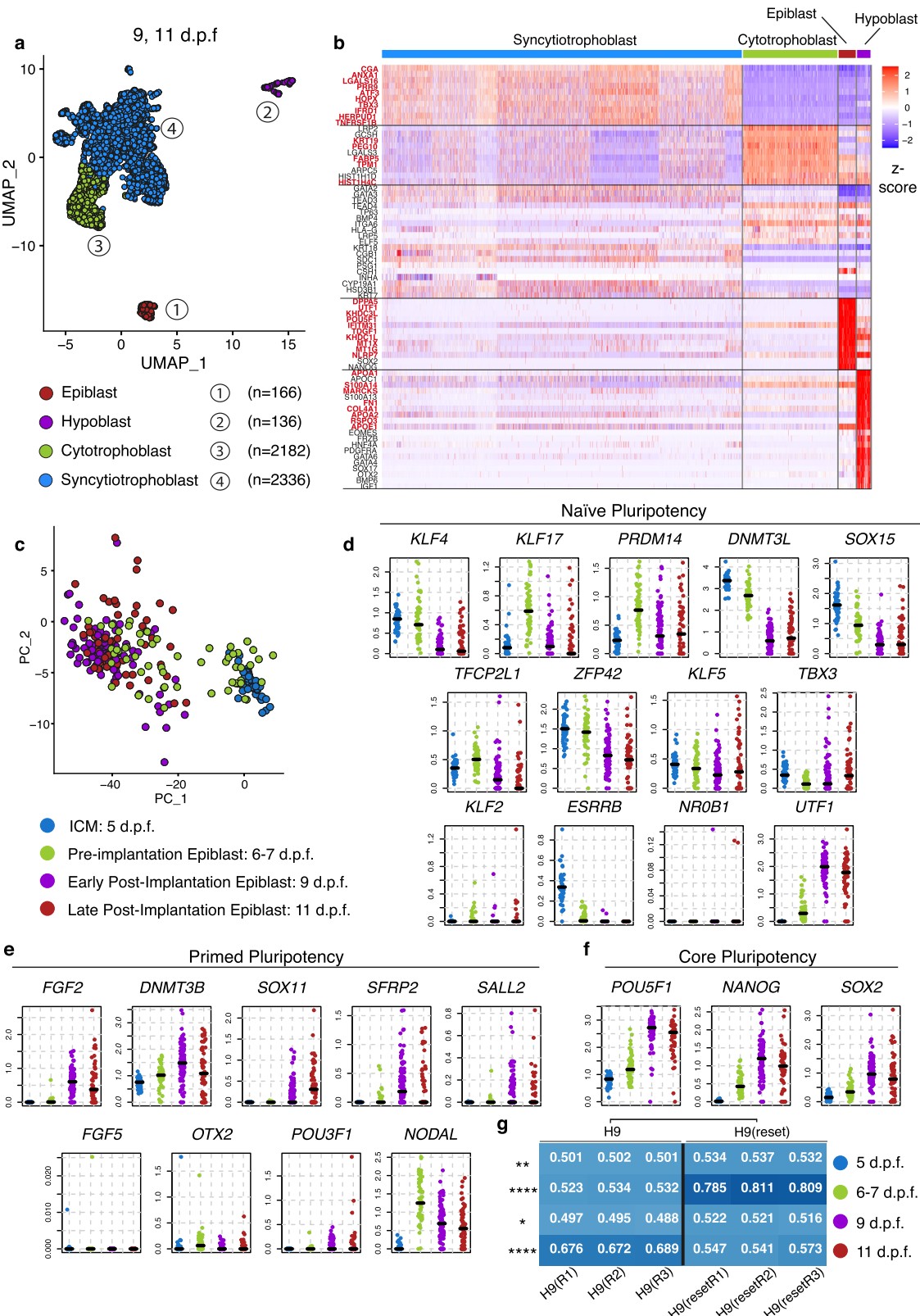

*NODAL* exhibited a downward trend during this developmental time-window. All the core pluripotency markers *POU5F1*, *NANOG* and *SOX2* showed a steady upregulation as the epiblast transited from pre- to post-implantation (Fig. 1f), in agreement with a recent scRNA-seq dataset[23] (Supplementary Fig. 4a, b).

A variety of protocols have been developed to derive human embryonic stem cells (ESCs) reflecting different pluripotency states[3,39–42]. To address whether these different cell types correspond to different stages of embryo development, we used logistic regression analysis to project human ESCs cultured

**Fig. 1 Single cell RNA sequencing of post-implantation human embryos and pluripotency transition profile. a** Identification of four major clusters coloured by lineage via UMAP: epiblast (red, 108 cells at 9 d.p.f., 58 cells at 11 d.p.f., total=166), hypoblast (purple, 82 cells at 9 d.p.f., 54 cells at 11 d.p.f., total = 136), cytotrophoblast (green, 1956 cells at 9 d.p.f, 226 cells at 11 d.p.f, total=2182), syncytiotrophoblast (cyan, 1058 cells 9 d.p.f., 1278 cells at 11 d.p.f., total = 2336). Number of embryos: n = 8 at 9 d.p.f., $n$ = 8 at 11 d.p.f. Only embryos with all three lineages were analysed, refer to Supplementary Data 1. **b** Heatmap displaying z-score values of the most differentially expressed genes between different lineages, alongside known lineage markers. In bold red the top most enriched genes expressed in a particular lineage as shown in Supplementary Data 2. **c** Comparisons between pre-implantation single-cell sequenced epiblast and ICM cells previously published[33-36] (cyan inner cell mass (ICM) at 5 d.p.f., green epiblast at 6–7 d.p.f.) versus our post-implantation datasets (purple epiblast at 9 d.p.f, red epiblast at 11 d.p.f.). Comparison of the expression levels of key markers of naïve pluripotency (**d**), primed pluripotency (**e**) and core pluripotency (**f**). Bars represent median. **g** Logistic regression analysis showing quantitative cell matching scores between human embryo datasets ICM at 5 d.p.f. (blue), pre-Epi at 6–7 d.p.f. (green) and post-Epi at 9 d.p.f. (purple) and at 11 d.p.f. (red) (y-axis) and human embryonic stem cells H9 and H9-Reset[43] (x-axis). Comparison H9 vs H9-Reset by Sidak's multiple comparisons ad-hoc test for 2 way ANOVA, H9 cells differ from the H9-Reset throughout stages: 5 d.p.f. ($p$ = 0.0015**), 6–7 d.p.f. ($p$ < 0.0001****), 9 d.p.f. ($p$ = 0.0101*) and 11 d.p.f. ($p$ < 0.0001****). This difference is highly significant at 6-7 d.p.f., with H9-Reset cells sharing similarities to 6–7 d.p.f. pre-implantation epiblast, and at 11 d.p.f. with H9 cells sharing similarities with 11 d.p.f. post-implantion epiblast.

under different regimes onto the pre- and post-implantation epiblast[43–45] (Fig. 1g, Supplementary Fig. 5a, b). Despite similarities with both pre and post-implantation stages, conventional primed ESCs (H9) shared transcriptional similarities with the post-implantation human epiblast at 11 d.p.f., while recently generated naïve human ESCs[43] (H9-Reset) more closely resembled the pre-implantation epiblast at 6–7 d.p.f. (Fig. 1g). When these cells were induced to exit naïve pluripotency and became "capacitated" for germ layer specification[45], their gene expression profile changed from a pre-implantation-like to a post-implantation-like epiblast gene signature (Supplementary Fig. 5b). Overall, our data indicate a progressive pluripotency transition of the epiblast following implantation, in agreement with recent studies of human embryos[23,46], and expression profiles observed in mouse and monkey embryos[7,8,47].

**FGF signalling is required for embryo growth after implantation.** Among the primed pluripotency markers upregulated by the post-implantation human epiblast (Fig. 1e), *FGF2* represents an important ligand of the fibroblast growth factor (FGF) signalling cascade as activation of this pathway is required for derivation and maintenance of human ESC in culture[48]. Analysis of the expression of the major ligands and receptors of the FGF family (Supplementary Data 5) revealed the post-implantation epiblast as a central source of two major transcripts: *FGF2* and *FGF4* (Fig. 2a). Epiblast cells expressed low levels of additional FGF ligands such as *FGF8, FGF17, FGF18* and *FGF19* (Supplementary Fig. 6a–d). The corresponding FGF receptors (*FGFR1/2/3/4*) were widely expressed in all post-implantation lineages (Fig. 2b), with particular enrichment of *FGFR1* in the hypoblast, suggesting possible signalling interactions.

To test the functional role of FGF signalling, we cultured blastocysts from 6 to 8 d.p.f. (48 h) in the presence of one of three agents: PD-0325901 (PD), which inhibits MEK downstream of the FGFR; the pan-FGFR inhibitor, LY2874455;[49] or a combination of FGF2, FGF4 and heparan sulphate to stimulate activation of the FGF signalling cascade. To determine whether any change in cell numbers was due to inhibitor action or non-specific cytotoxic effects, embryos were exposed to two different concentrations of inhibitors: PD at high (3 μM) and lower dose (1 μM) as previously described[7,50], and LY at high (1 μM), as shown to be able to suppress downstream FGF targets[51], and lower dose (500 nM)[52]. Either inhibition of MEK by PD or blockage of the upstream FGF receptors by LY caused a reduction in cell numbers in both the epiblast and hypoblast lineages (Fig. 2c–f and Supplementary Fig. 6e, Supplementary Data 6). While this was observed at the lower concentration of LY, supplementation of PD at 1 μM had only a mild effect in the reduction of the number of OCT4-positive cells compared to

control (non-significant). However, supplementation of 1 μM PD led to a significant reduction of hypoblast cells compared to control. In contrast to the epiblast and hypoblast, the number of trophoblast cells was not affected by PD. Because blockage of all FGFRs by LY caused a reduction in the number of trophoblast cells, alongside epiblast and hypoblast, we cannot rule out a potential toxic effect of the drug itself in addition to the effect of removing FGF signalling entirely. Despite the reduction in cell number, embryos treated with LY were able to attach and progress in their development. These results suggest that FGF signalling is necessary for proliferation of the epiblast, hypoblast and trophoblast lineages, even though we cannot rule out that the reduction of hypoblast cells observed could be due to the reduction of epiblast cells and thereby an indirect effect of FGF inhibition. The reduction of cell numbers by the upstream inhibition through LY, which was not replicated when inhibiting the MEK/ERK downstream signalling at the lower inhibitor concentration, suggest the activity of additional pathways downstream of FGF. We also tested the effect of FGF signalling activation by addition of exogenous FGF2 and FGF4 to the culture medium. Treated embryos were comparable in cell number to controls (Fig. 2 d–f and Supplementary Fig. 6f) but displayed a reduced variability in the epiblast cell numbers (Supplementary Fig. 6g), supporting the possibility that FGF may be beneficial for the development of human embryos in culture.

**Molecular characterisation of anterior-like hypoblast cells.** Another major step of post-implantation development involves pre-patterning of the epiblast before initiation of gastrulation, which is achieved by the AVE[15] via secretion of NODAL, BMP and WNT antagonists[53,54]. The region of the epiblast in proximity to the AVE becomes the future anterior side of the embryo, while the opposing region becomes the site of the primitive streak formation. Therefore, we investigated whether a similar signalling centre may exist in the human hypoblast. Analysis of the single-cell transcriptomes of the hypoblast lineage identified a sub-cluster of cells enriched for the expression of *CER1* both at 9 and 11 d.p.f. (sub-cluster 0; Fig. 3a, b and Supplementary Fig. 7a, b), a key NODAL/BMP/WNT antagonist secreted by the mouse[55] and monkey[11] AVE. Immunofluorescence analysis at 9 d.p.f. (Fig. 3c, d) confirmed that 17-43% (IQR) of the GATA6-positive hypoblast cells in contact with the epiblast expressed CER1 (Fig. 3e–g, Supplementary Data 7). These CER1 positive cells showed a localisation bias towards one side of the hypoblast in a subset of embryos (Fig. 3h, Supplementary Fig. 7d–f, Supplementary Video 1). These results suggest that a subpopulation of cells within the human hypoblast may act similarly to the mouse AVE by secreting the antagonist CER1.

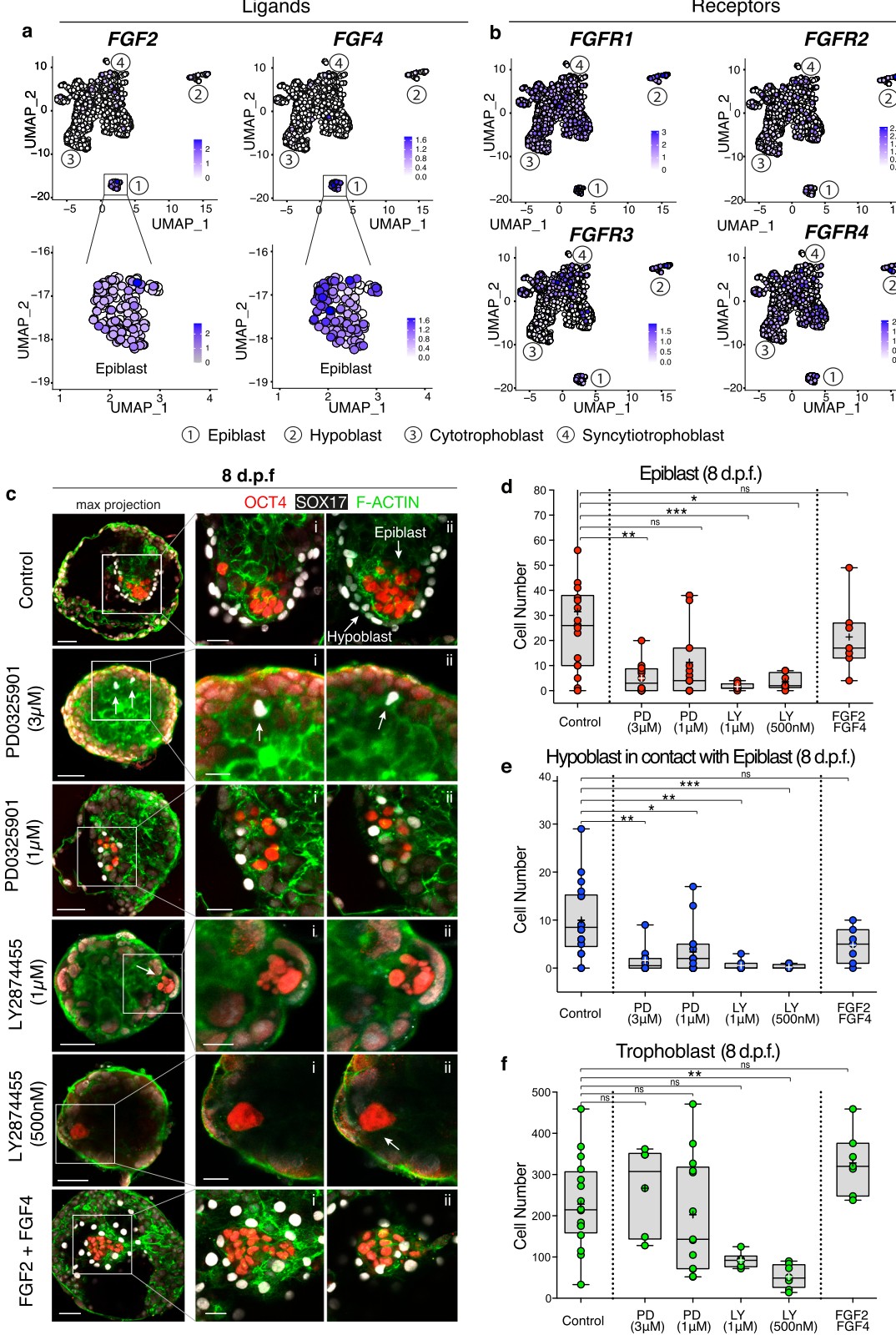

① Epiblast  ② Hypoblast  ③ Cytotrophoblast  ④ Syncytiotrophoblast

Given the localisation bias of CER1, we sought to investigate which upstream transcription factors may drive hypoblast patterning. Using the co-expression and motif-supported pipeline SCENIC[56], we identified 12 transcription factors predicted to participate in regulation of *CER1* expression based on co-expression networks and the presence of a binding motif

upstream of the *CER1* coding sequence (Supplementary Fig. 8a). The activity of each of these transcription factors was scored by SCENIC based on the expression of its putative targets in every single cell, producing a regulon activity score as a proxy for predicted transcription factor activity (Supplementary Fig. 8b-m; Red UMAPs). These factors included the transcription factors

**Fig. 2 FGF signalling is required for proliferation during post-implantation development. a/b** UMAPs showing expression levels of main FGF ligands (*FGF2/4*) (**a**) and receptors (*FGFR1/2/3/4*) (**b**). The epiblast is a source of *FGF2/4*. FGF receptors (*FGFR1/2/3/4*) are widely expressed throughout all the lineages, with *FGFR1* being the most highly expressed. **c** Immunofluorescence of human embryos at 8 d.p.f. cultured in DMSO control, in MEK inhibitor PD0325901 at 3 and 1 μM, in pan-FGFR inhibitor LY2874455 at 1 μM and 500 nM, in FGF2/4+heparan sulphate proteoglycans. On the left: zoom-out maximum projection, on the right z-sections. Number of experimental replicates: 9. **d–f** Quantification of the cell number for the epiblast marked by OCT4 (**d**), hypoblast by SOX17 expressing cells in contact with the basal side of epiblast (**e**), and trophoblast (negative for OCT4/SOX17) (**f**). Statistical test: two-tailed Kruskal–Wallis test with Dunn's correction. **d** Inhibition by PD 3 μM causes significant reduction of OCT4 cells, non-significant at 1 μM: $p = 0.0026$ (**) control vs PD 3 μM. Inhibition by LY causes a significant reduction both at 1 μM and 500 nM: $p = 0.0006$ (***) control vs LY 1 μM, $p = 0.0109$ (*) control vs LY 500 nM. Non-significant difference when supplemented with FGF2/4. **e** Inhibition by PD causes a significant reduction of SOX17 cells both at 3 μM and 1 μM: $p = 0.0023$ (**) control vs PD 3 μM, $p = 0.0203$ (*) control vs PD 1 μM. Inhibition by LY causes a significant reduction both at 1 μM and 500 nM: $p = 0.0011$ (**) control vs LY 1 μM, $p = 0.0002$ (***) control vs LY 500 nM. Non-significant difference when supplemented with FGF2/4. **f** Inhibition by PD does not cause a significant reduction in trophoblast cells, either at 3 μM or 1 μM. Inhibition by LY causes significant reduction only at 500 nM: $p = 0.0015$ (**) control vs LY. Non-significant difference when supplemented with FGF2/4. Box represents the 25th–75th percentiles interval, middle line the median, cross represents the mean, whiskers show the minimum and maximum values, dots represent individual embryos. Number of embryos: control ($n = 19$), PD 3 μM ($n = 12$), PD 1 μM ($n = 15$), LY 1 μM ($n = 8$), LY 500 nM ($n = 8$), FGF2/4 ($n = 7$). Scale bars: 50 μm (Fig. 2c) and 25 μm (Fig. 2c i–ii, zoom). Source data are provided as a Source Data file.

FOXA2, HNF4A, JUNB, SP5 and GSC, which did not show preferential activity in the CER1-positive hypoblast sub-cluster (Supplementary Fig. 8b–g, Supplementary Data 9), while the mouse AVE marker HHEX, as well as FOXA3, LHX1, MIXL1, MNX1 and SOX21 did show preferential activity in the putative anterior hypoblast sub-cluster (sub-cluster 0) (Supplementary Fig. 8h–m, Supplementary Data 9).

To define the gene expression profile of this sub-population in the hypoblast, we examined the most enriched genes that characterise the sub-cluster 0 (Supplementary Fig. 9a) and analysed the correlation between *CER1* expressing cells and other canonical AVE markers[57]. *CER1* expression correlated strongly with the expression of two major NODAL antagonists, *LEFTY1 and LEFTY2* both at 9 and 11 d.p.f., which were also found to be among the most enriched genes of this sub-cluster, and to a lower extent with the BMP antagonist *NOG*, the WNT antagonists *DKK1/4* and *HHEX* (Fig. 3i and Supplementary Fig. 9b–d), that are expressed in the mouse AVE[58], demonstrating the co-expression of multiple canonical AVE markers at the RNA level in the hypoblast. We confirmed that cells secreting LEFTY1 protein at 9 d.p.f. also displayed a localisation bias towards one side of the hypoblast in a subset of embryos (Fig. 3j–l, Supplementary Fig. 10a), similar to CER1 secreting cells.

**Patterning of the epiblast by anterior-like hypoblast.** To determine whether this group of human hypoblast cells expressing anterior markers is able to exert a similar action as the mouse AVE in pre-patterning the epiblast before gastrulation, we next investigated whether canonical targets of signalling pathways were heterogeneously expressed in the epiblast. We found heterogeneous expression of the WNT target *AXIN2* (Supplementary Fig. 11a) as well as of the WNT nuclear effector *TCF7* (*TCF1*) (Supplementary Fig. 11b). By analysing the BMP targets *ID1*, *ID2*, *ID3* and *ID4* (Supplementary Fig. 11c–f), we observed heterogeneity across the epiblast and restricted expression of *ID2* in a subset of cells. Cells expressing *ID2* corresponded to a small number of cells that showed both SMAD1 (BMP) and SMAD3 (NODAL/TGFβ factor) transcriptional activity (Supplementary Fig. 11g, h). Cells with active TCF7 and SMAD1 showed overlap, suggesting that WNT and BMP signalling are active simultaneously in the same cells (Supplementary Fig 11i).

To validate the heterogeneous signalling across the epiblast and visualise its localisation, we investigated the relationship between CER1 localisation and the activation of the BMP pathway as marked by the presence of phosphorylated (p)SMAD1.5 (Fig. 3m). Nuclear pSMAD1.5 was visible in a subset of epiblast cells (arrows) distanced from CER1 positive hypoblast cells (Fig. 3m,

Supplementary Fig. 10b), whereas OCT4 epiblast cells adjacent to CER1 positive hypoblast cells were negative for pSMAD1.5. Together, these data provide support for the existence of a subpopulation of cells within the human hypoblast that resembles the mouse AVE and secrets conserved signalling antagonists for patterning of the epiblast. However, rather than being organised into a discrete domain as in the mouse embryo[57], in human this group of cells occupies a larger domain preferentially distributed along one side of the hypoblast. We term these cells the human putative anterior hypoblast signalling centre.

**Establishment of anterior-like hypoblast at 7 d.p.f.** The establishment of the AVE in the mouse is preceded by the formation of the DVE at the distal tip of the epiblast[53,54,59]. The anterior side becomes specified through the migration of DVE cells towards the proximal pole of the embryo[14,60]. To investigate the dynamics driving the formation of the putative human anterior hypoblast over time, we examined the distribution of CER1-expressing cells upon attachment of the embryo at 7 d.p.f. 69–82% (IQR) of the GATA6-positive hypoblast cells in contact with the epiblast expressed CER1 (Fig. 4a–c, Supplementary Data 10), as opposed to 17–43% in embryos at 9 d.p.f. (Fig. 3g). The CER1 expressing cells occupied a widespread domain across the hypoblast at 7 d.p.f. (Fig. 4d and Supplementary Fig. 12a, b). This is in contrast to the discrete domain of the mouse DVE and appears to resemble, instead, the pattern observed in the hypoblast of cynomolgus monkey[11] and chick embryos[61,62]. Examination of their angular distribution revealed that CER1 expressing cells at 7 d.p.f. did not show any localisation bias (average cellular distribution of 91°) (Fig. 4e, Supplementary Fig. 13a–c), whereas the position of the CER1 domain became shifted towards one side of the epiblast at 9 d.p.f. (average 71°) (Supplementary Fig. 13d). These data suggest a model where CER1 occupies a significantly wider domain of the hypoblast at 7 d.p.f. and becomes asymmetrically localised to one side at 9 d.p.f. (Fig. 4f).

**Discussion**
Our large scRNA-seq dataset of human embryos generated in this study allowed us to investigate key signalling events underlying human post-implantation development. We first focused on the pluripotency state transition of the human epiblast as it develops beyond implantation. This revealed that the downregulation of naïve genes is accompanied by the upregulation of early post-implantation primed factors, in agreement with previous results[23,24]. This pluripotency transition is needed for epiblast epithelialisation, as we have previously shown[32], and is recapitulated

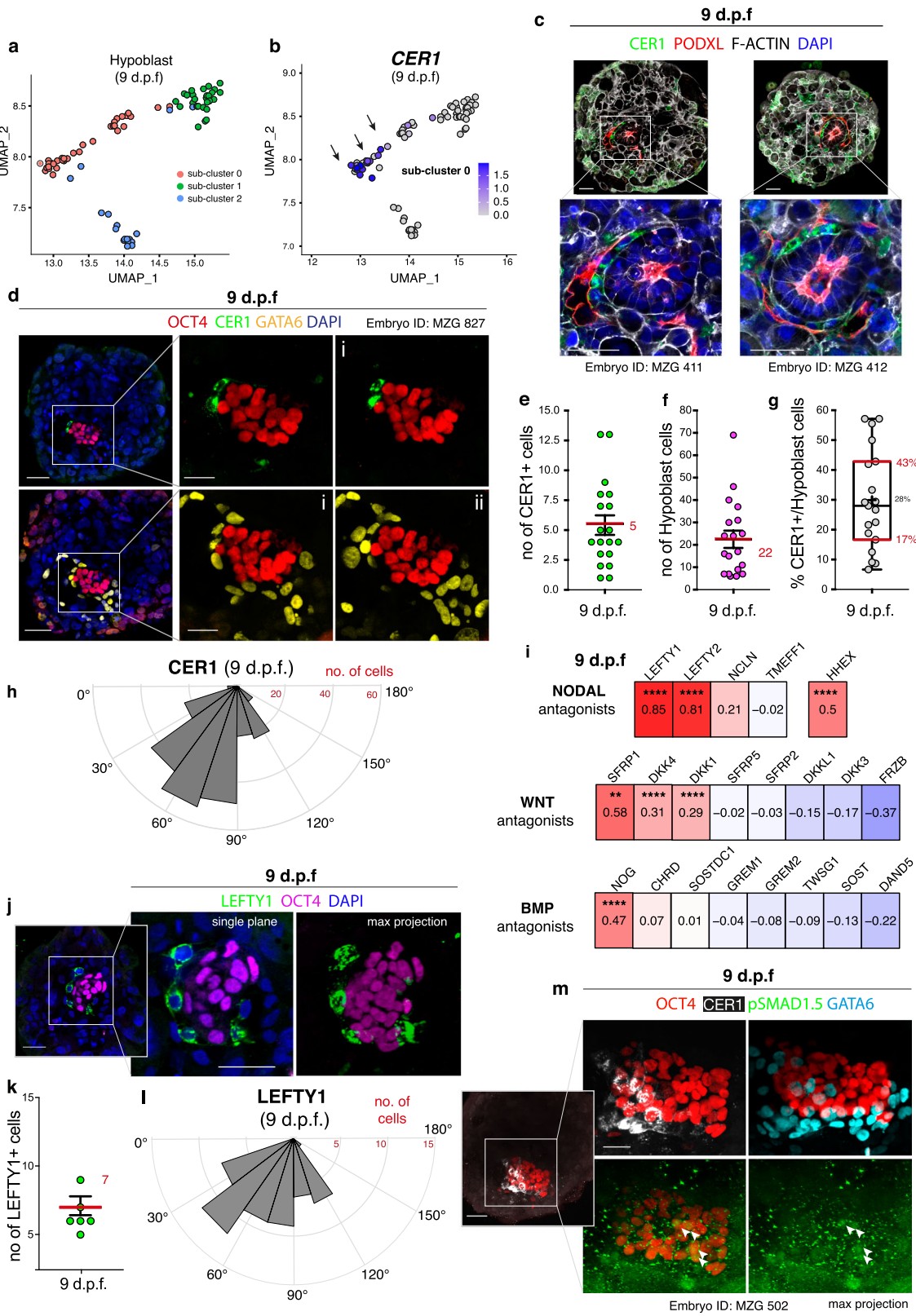

by different pluripotent states of human ESCs in culture[3]. Interestingly, no cells were similar to the peri-implantation epiblast suggesting a possible intermediate pluripotent state that is difficult to capture[25]. Such a state could potentially correspond to a 'rosette stage' recently described in mESCs[63].

In contrast to the specification of the primitive endoderm in mouse[64–66], FGF/ERK signalling has been shown to be dispensable for the lineage segregation between epiblast and hypoblast in the human embryo[50,67]. However, here we find that FGF signals are necessary for growth and proliferation of the epiblast,

**Fig. 3 Characterisation of the putative anterior hypoblast in human embryos at 9 d.p.f. a** UMAP of hypoblast cells subdivided into sub-clusters 0-2. Most enriched genes expressed in each sub-cluster are reported in Supplementary Data 8. **b** Sub-cluster 0 expresses *CER1*. **c, d** Immunofluorescence at 9 d.p.f.: CER1 is expressed asymmetrically in a subset of hypoblast cells (GATA6+). N (experimental replicates): 3. N(embryos): 19 **e**, Quantification CER1+ cells ($n = 19$ embryos, mean in red ± SEM). **f** Quantification hypoblast cells (GATA6+) in contact with basal epiblast ($n = 19$ embryos, mean in red ±SEM). **g** Percentage of CER1+ cells versus total hypoblast cells per embryo. Boxes represent the 25th/75th percentiles, red line the median, cross the mean, whiskers the min/max, dots individual embryos (n = 19); N(experimental replicates): 3. Source data provided as a Source Data file. **h** Quantification of angular distribution of CER1+ cells along the hypoblast hemisphere (0° to 180°). N(embryos) $n = 28$, combined from Supplementary Figure 7f (all embryos included). CER1+ cells show a significant localisation bias towards one side of the hypoblast in 10/28 embryos (Supplementary Figure 7f). **i**, Correlation analysis of *CER1* with WNT, BMP and NODAL antagonists. Co-expression corrected by Benjamini–Hochberg: significant correlations between *CER1* and *LEFTY1* ($p = 4.21E^{-13}$), *LEFTY2* ($p = 2.75E^{-10}$), *HHEX* ($p = 1.88E^{-07}$), *NOG* ($p = 1.39E^{-08}$), *DKK4* ($p = 4.41E^{-07}$), *DKK1* ($p = 1.65E^{-05}$), *SFRP1* ($p = 0.0047$). Correlation with *NCLN*, *CHRD* and *SOSTDC1* are ns. **j** Immunofluorescence of LEFTY1 at 9 d.p.f.. N(experimental replicates): 3. N(embryos): 7. **k** Quantification of LEFTY1+ cells ($n = 7$ embryos, mean in red ± SEM). **l** Quantification of angular distribution of LEFTY1+ cells along the hypoblast hemisphere. N(embryos) $n = 7$, combined from Supplementary Fig. 10a. LEFTY1+ cells showed a statistically significant localisation bias in 2/7 embryos (Supplementary Figure 10a). **m** Immunofluorescence at 9 d.p.f. of nuclear pSMAD1.5 in a subset of OCT4+ cells (arrows) distant from CER1 domain, where pSMAD1.5 is not detected. N(experimental replicates): 8. N(embryos) stained for pSMAD1.5 = 29; 9/29 embryos (31%) show pSMAD1.5 in OCT4+ cells; 8/9 embryos (89%) display localisation of pSMAD1.5 distant to CER1. Scale bars: 50 µm (Fig. 3c,j, Fig. 3d,m (left)), 25 µm (Fig. 3d,m (right)). Source data are provided as a Source Data file.

hypoblast and trophoblast during early post-implantation development. We cannot exclude that FGF may play also a role in fate specification as complete lineage segregation between the epiblast and hypoblast occurs only upon implantation[68]. In agreement with this, a recent study reported that FGF signalling is required for the differentiation of naïve human ESCs to a hypoblast-like fate[46,69]. In addition to ERK, FGF signalling may also activate the AKT cascade, which is necessary for human embryo pre-implantation development[70]. Supplementation of IGF1 to activate the AKT cascade has been suggested as a potential replacement of FGF supplementation for human ESC culture[70]. Given the use of IGF1 in our culture media, the decrease in the epiblast cell numbers variability when supplementing additional FGF, and the strong effect of both FGF inhibitors, we conclude that IGF1 and FGF signalling may have intersecting rather than redundant roles.

The results we present here identify the existence of a group of hypoblast cells that express antagonists of the BMP, NODAL and WNT signalling pathways, such as CER1 and LEFTY1 (Fig. 4f), which may induce pre-patterning of the epiblast to establish the AP axis of the human embryo. This step is of fundamental relevance for the initiation of gastrulation, as the primitive streak will form at the site distal from this putative signalling centre. Future studies will be required to dissect the mechanism by which this putative anterior signalling centre patterns the epiblast and regulates establishment of the AP axis. However, in contrast to the mouse where the DVE is established at the distal pole of the egg cylinder as a discrete domain[53,54,59], the pattern of gene expression in humans occupies a widespread domain at 7 d.p.f., reminiscent of the pattern observed in the hypoblast of cynomolgus monkey[11] and chick embryos[61,62]. It would be important in future studies to determine the mechanism underlying the shift in localisation towards one side of the epiblast at 9 d.p.f.. This could occur as a result of a local downregulation of CER1 from an initially widespread expression, causing only one side of the hypoblast to retain its expression, or, alternatively, be a consequence of the uneven expansion of GATA6+/CER1- cells in comparison to GATA6+/CER1+ cells.

While our data highlights the possible function of the hypoblast in mediating epiblast patterning via secreted inhibitors, the role of the trophoblast and amnion as a source of BMP and WNT signals remains to be explored. In cynomolgus monkey embryos the cytotrophoblast secretes WNT3a while the amnion acts as a source of both BMP4 and WNT3a[11]. Interestingly, in humans WNT3a is not highly expressed in the cytotrophoblast as in the monkey, raising further questions of the role and action of

different WNTs in the human specific context. Overall, our study points to signalling interactions in the developing human embryo leading up to gastrulation.

## Methods

**Ethics statement**. Human embryo work was done in accordance with Human Fertility and Embryology Authority (HFEA) regulations (license reference R0193). Ethical approval was obtained from the 'Human Biology Research Ethics Committee' of the University of Cambridge (reference HBREC.2017.21). Informed consent was obtained from all participants in the study. These were patients from the CARE Fertility Group, Bourn Hall and Herts & Essex fertility clinics, which had supernumerary embryos after completing their IVF treatment. Prior to giving consent, patients were informed about the specific objectives of the project, and the conditions that apply within the license. They were also offered counselling and did not receive any financial inducements. Embryo cultures were always stopped before day 14 and prior to the appearance of any signs of primitive streak formation.

**Human embryo thawing**. Cryopreserved day 5 or day 6 human blastocysts were received at the University of Cambridge and thawed using Kitazato thaw kit (VT802-2, Hunter Scientific) following the manufacturer's instructions. The day before thawing, the TS (Thawing) solution was placed at 37 °C and Global Total human embryo culture medium (HGGT-030, LifeGlobal group) was incubated at 37 °C + 5% $CO_2$. Upon thawing, human embryos were immersed in 1 mL of the pre-warmed TS solution for 1 min. They were subsequently transferred to DS (Dilution) solution for 3 min, WS1 (Washing step 1) solution for 5 min and WS2 (Washing step 2) solution for 1 min. All these incubation steps were done using reproplates (REPROPLATE, Hunter Scientific). Thawed embryos were incubated in pre-equilibrated Global Total human embryo culture medium under mineral oil (9305, Irvine Scientific) for 4 h to allow recovery. Culture conditions are the following: 37 °C 20% $O_2$ and 5% $CO_2$ in a humidified atmosphere.

**Post-implantation culture of human embryos**. 4 h following the thawing, the *zona pellucida* was removed by briefly treating the embryos with acidic Tyrode's solution (T1788, Sigma). Embryos were cultured in pre-equilibrated in vitro culture 1 (IVC1) culture medium (M11-25, Cell Guidance Systems). We introduced a modification to our original IVC1 recipe[6] consisting of the addition of 50 ng/mL of Insulin Growth Factor 1 (IGF-1) (78078, STEMCELL Technologies), as this improved the preservation of epiblast and hypoblast cells in culture. Embryos were cultured in 8 well µ-slides tissue culture wells (80826, Ibidi) in approximately 300 µL volume per embryo per well. Half-media changes with IVC1 supplemented with IGF-1 were done every 24 h.

**Limitations of the culture system and exclusion criteria for the analysis**. In this study we used supernumerary embryos donated from couples undergoing IVF treatment. As a consequence, several embryos degenerated upon thawing or during the culture before reaching the desired final stage. These were usually unable to attach to the dish, showed clear signs of cell death and were therefore excluded from the analysis. Regarding the embryos included in the analysis, it is important to consider the variability in sample quality that exists among different embryos and patients. This should be taken into consideration for the interpretation of the data of the study. Another source of variability is the culture condition. This warrants further optimisation to support optimal development of the human embryo post-implantation which is currently lacking.

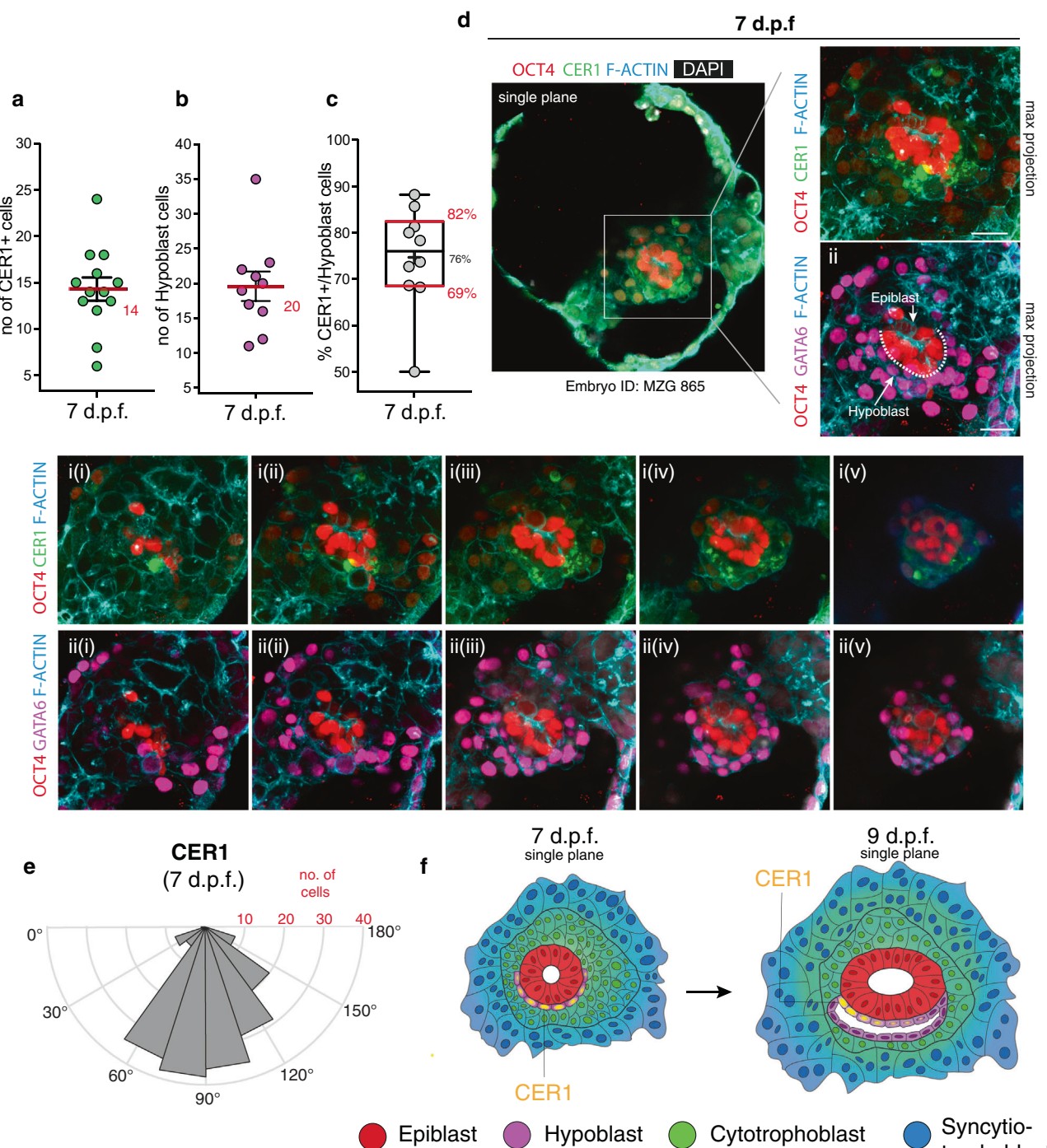

**Fig. 4 Characterisation of the putative anterior hypoblast in the human embryos at 7 d.p.f. a** Quantification of the number of CER1 expressing cells ($n = 13$ embryos, mean in red ± SEM). **b** Quantification of the number of hypoblast cells as GATA6+ cells lining the basal side of epiblast ($n = 10$ embryos, mean in red ±SEM). **c** Percentage of hypoblast cell expressing CER1. Boxes represent the 25th and 75th percentiles, red line the median, cross the mean, whiskers the min and max values, dots individual embryos ($n = 10$ embryos). N experimental replicates: 5. **d** Immunofluorescence analysis of embryos at 7 d.p.f. reveals the widespread distribution of CER1 cells along the hypoblast, marked by GATA6 along the basal side of the epiblast epithelium (ii). N experimental replicates: 5. N(embryos) analysed: 13 **e** Quantification in 3D of the angular distribution of CER1 expressing cells along the hypoblast hemisphere at 7 d.p.f. (0° to 180°). N embryos: 10. **f** Model of the establishment of the putative anterior hypoblast centre: initially at 7.d.p.f. the putative anterior hypoblast centre is radially expressed throughout most of the cells of the hypoblast and particularly enriched distally. At 9.d.p.f. the putative anterior hypoblast centre becomes localised asymmetrically to one side of the hypoblast, repressing the activity of key signalling pathways such as BMP on the adjacent epiblast. Asymmetric activity may discriminate the future anterior region from the prospective posterior region of the epiblast, where gastrulation will occur. Scale bars: 50 µm (4a, zoom out), 25 µm (4d, i–xii). Source data are provided as a Source Data file.

**Inhibitor treatments**. For the analysis presented in Fig. 2, the culture medium has been supplemented with DMSO for control group; with 3 μM or 1 μM PD0325901 (PD, Stem Cell Institute, Cambridge) as reported in other studies of MEK inhibition;[50] with LY287445 (ab216313, Abcam) to inhibit all FGF receptors, at 1 μM, as shown to be able to inhibit pERK[51], and at 500 nM[52]. To stimulate FGF signalling, the medium has been supplemented with FGF2 (Stem Cell Institute) at 25 ng/ml, FGF4 (R&D systems, 67-202) at 25 ng/ml and Heparin (Sigma H3149) at 1×. Apart from degeneration during the culture as discussed above, all embryos reaching stage 8 d.p.f. have been included in the analysis presented in Fig. 2. For all the other experiments presented, only embryos with clear epiblast and hypoblast were included in the analysis. Embryos, which have successfully attached by the end of the culture but lack either epiblast or hypoblast or show clear signs of degeneration have been excluded and are not accounted for the n number reported in this study.

**Single cell isolation from in vitro cultured human embryos**. Embryos were cultured from 5 d.p.f. as above until either 9 or 11 d.p.f.. At the end of the culture, each embryo was gently detached from the bottom of the ibidi slide using a thin plastic needle and incubated in 500 μL of pre-equilibrated 0.5% Trypsin EDTA (25300054, Life Technologies) at 37 °C for 10 min. The reaction was stopped by transferring the embryos in PBS + 1% PVP (polyvinylpyrrolidone) or 0.1% BSA. The embryo was dissociated into single cells by mechanical pipetting using 50 μm diameter tips (MXL3-IND-50, Origio). The single cell suspension was loaded in the 10x-Genomics Chromium for encapsulation. The 10x-Genomics v2 or v3.0 libraries were prepared as per the manufacturer's instructions. Embryos labelled as 'mplex' in Supplementary Data 1 were multiplexed under the same reaction. Libraries were sequenced, aiming at a minimum coverage of 50,000 raw reads per cell, on an Illumina HiSeq 4000 (paired-end; read 1: 26 cycles; i7 index: 8 cycles, i5 index: 0 cycles; read 2: 98 cycles).

**Single cell RNA-seq data analysis**. Single-cell RNA-sequencing was performed using the 10X Genomics Chromium system. Reads were aligned against GRCh38 and transcript abundances counted using the Cell Ranger software suite (version 2.2). Cells for which expression of mitochondrial genes accounted for more than 20% of total expression or cells with fewer than 1000 detectable genes were removed from the dataset (Supplementary Fig. 1f–h). A total of 29 embryos was submitted for scRNAseq (Supplementary Data 1). We used strict filtering criteria: only 16 out of 29 embryos were included for analysis based on the presence of all three lineages (epiblast, hypoblast and trophoblast) to ensure viability of the embryos and preservation of signalling crosstalk between lineages. The final analysis includes 16 embryos: 8 embryos at 9 d.p.f. and 8 embryos at 11 d.p.f. as presented in Supplementary Data 1. Embryos that were sequenced in a multiplexed fashion were computationally demultiplexed using the souporcell algorithm[71], which clusters the cells by an inferred genotype without the need for a reference genotype.

Further downstream analyses were performed in the R package Seurat[72](version 3.1.5). Batch correction relied on identification of integration anchors[73]. Cell clusters were identified using a shared nearest neighbour graph approach based on the aligned components, as implemented in the 'FindClusters' function. Within this function, the resolution parameter was set such that the resulting clustering captured most of the biological variance, without over-splitting the data. This amounted to a resolution parameter of 0.05. Single-cell data was further visualised using the UMAP dimensionality reduction, as determined by the 'RunUMAP' function in Seurat. Cluster-defining genes were identified using the FindAllMarkers function.

Data from Takashima et al.[43] was aligned to GRCh38 using kallisto[74]. Integrated data on ICM and epiblast cells from Stirparo et al.[33] was subjected to the same workflow within Seurat as described before to integrate with our data. Data from Xiang et al.[23] and Zhou et al.[24], was subjected to the same workflow in Seurat as described above to visualise gene expression. Cell line data from Takashima et al.[43], Theunissen et al.[75] and Rostovskaya et al.[45] were mapped onto the epiblast clusters using a logistic regression framework[74]. Embryo clusters from Xiang et al.[23], and Zhou et al.[24] were mapped onto our embryo clusters using the same logistic regression framework. To identify transcription factor regulons and score their activity in single cells as presented in Supplementary Fig. 8, the SCENIC pipeline was employed with default settings. The area under the curve score for each regulon was then used to create a new assay in the Seurat object for visualisation of specific regulon activity on the previously generated UMAP.

**Embryo fixation and immunostaining**. Human embryos were fixed at day 9 of development in 4% paraformaldehyde (11586711, Electron Microscopy Sciences) for 20 min. They were permeabilised in PBS containing 0.3% Triton X-100 and 0.1 M glycine (permeabilisation buffer) for 30 min at room temperature. Samples were blocked for 1 h in PBS containing 1% bovine serum albumin (BSA) and 0.1% Tween and incubated with primary antibodies diluted in blocking buffer. Fluorescently conjugated Alexa Fluor secondary antibodies (ThermoFisher Scientific) were incubated for 2 h at room temperature in blocking buffer. The following primary antibodies were used: goat polyclonal anti CERBERUS1 (AF1075, R&D Systems, 1/200), goat polyclonal anti GATA6 (AF1700, R&D Systems, dilution

1/200), goat polyclonal anti LEFTY1 (AF746, R&D Systems, dilution 1/200), mouse monoclonal anti OCT3/4 (sc-5279, Santa Cruz Biotechnology, dilution 1/200), mouse monoclonal anti PODOCALYXIN (MAB1658, clone 222328, R&D Systems, dilution 1/500), rabbit monoclonal anti Phospho-Smad1 (Ser463/465)/ Smad5 (Ser463/465)/ Smad9 (Ser465/467) (D5B10, Cell Signalling Technology, dilution 1/100) and goat polyclonal anti SOX17 (AF1924, R&D Systems, dilution 1/200). Where indicated secondary antibodies were incubated together with Alexa Fluor 488/594 phalloidin (A12379, ThermoFisher Scientific, dilution 1/500) and DAPI (D3571, ThermoFisher Scientific, dilution 1/1000).

For pSmad1.5 staining embryos were fixed for 15 min in 4% PFA, washed 3× in PBST, and subsequently incubated in 100% MeOH for 10 min. After rehydration in PBST, the embryos were permeabilised for 15 min in permeabilisation buffer. All solutions including blocking buffer were supplemented with cOmplete protease inhibitor (11836153001, Sigma Aldrich, dilution 1/1000).

**Image quantification and analysis**. Immunofluorescence images were processed and analysed using Fiji software[76] (http://fiji.sc). Cell number as presented in Fig. 3f–h and Fig. 4b–d was counted manually using the multi-point Cell Counter plugin in Fiji. To define the hypoblast, only GATA6+ cells lining and in direct contact with the basal distal side of the epiblast were counted, as opposed to the total GATA6 + cells, which includes parietal lineage. CER1 + cells in contact with the epiblast were counted with DAPI as nuclear reference. For each embryo, the number of GATA6 + hypoblast cells expressing CER1 was calculated as percentages as presented in box plots in Figs. 3g and 4c. The angle distribution of CER1 positive cells was quantified by using the 3D viewer of the SP8 Leica Lightning software as described in Supplementary Figs. 7 and 12. The epiblast was oriented in 3D in respect to the hypoblast to determine the proximal vs distal plane (where hypoblast is present). The angle for CER1 positive cells was determined by using the proximal plane and the middle point of the proximal surface of the epiblast as reference points. Angles are presented as frequency distributions (from 0° to 180°) plotted using Matlab software. To compare different embryos, 0° is taken as a reference point as the side where the majority of CER1 cells are localised along the entire hypoblast length (from 0° to 180°), defined by GATA6.

**Statistics and reproducibility**. The exact number of embryos sequenced is indicated in Supplementary Data 1. No statistical method was used to predetermine sample size. Embryos that did not expand after thawing were excluded from downstream analyses. The investigators were not blinded to group allocation during experiments and outcome assessment. For the analysis of immunofluorescence images, the normality of the data was determined using a D'Agostino Pearson normality test. Data not normally distributed were analysed by the non-parametric Kruskal–Wallis test with Dunn's correction. Data shown as boxes represent the 25th and 75th percentiles interval with the line in the middle of the box as the median, the cross represents the mean, whiskers show the minimum and maximum values, dots represent individual embryos. Statistical significance: $p < 0.05$ was considered statistically significant (*), $p < 0.01$ (**), $p < 0.001$ (***), $p < 0.0001$ (****). Statistical analyses were done in Prism (GraphPad) and R.

**Reporting summary**. Further information on research design is available in the Nature Research Reporting Summary linked to this article.

## Data availability

The raw sequencing and expression-count data with cell classifications have been deposited at the ArrayExpress database under accession code: E-MTAB-8060. Datasets can be visualised through the web portals www.humanembryo.org.

Regarding previously published datasets used here, single-cell RNA-sequencing data from Xiang et al.[23] and Zhou et al.[24] are deposited in Gene Expression Omnibus (GEO) under accession numbers GSE136447 and GSE109555, respectively. In addition, cell line data from Takashima et al.[43] are deposited at Array Express under accession number E-MTAB-2857. Cell line data from Theunissen et al.[75] are deposited at GEO under accession number GSE59435. Cell line data from Rostovskaya et al.[45] are deposited in GEO under accession number GSE 123055.

Immunofluorescence data will be available from the corresponding authors upon reasonable request. Source data are provided with this paper.

## Code availability

Custom scripts used for analyses have been deposited at https://github.com/TimCoorens/EarlyEmbryo_scRNA (https://doi.org/10.5281/zenodo.4738657)[77]. This repository is linked to Zenodo and citable as Molè et al. A single cell characterisation of human embryogenesis identifies pluripotency transitions and putative anterior hypoblast centre.

To align reads, reference assembly GRCh38 was used [https://www.ncbi.nlm.nih.gov/assembly/GCF_000001405.26/].

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

## Acknowledgements

We are grateful to all the patients donating their embryos, embryologists at the CARE, Bourn Hall and Herts & Essex clinics for help and support, and to colleagues in the M.Z.-G. laboratory. We thank Matthew Young for advice and help on data analysis and Krzysztof Polanski for helping with data sharing. We thank Connor Ross and the Jennifer Nichols group for the advice on using the LY inhibitor. We thank Martin Prete for creating the website. M.N.S. is funded by the European Molecular Biology Organisation (EMBO, Advanced EMBO fellowship) and UKRI Medical Research Council (MC_UP_1201/24). B.A.T.W. is funded by the Gates Cambridge Trust. Work in the laboratory of M.Z-G. is supported by grants from the Wellcome Trust (207415/Z/17/Z), Open Philanthropy/Silicon Valley, Curci and Weston Havens Foundations. S.B. is funded by the Wellcome Trust (Sanger core funding and personal fellowship to S.B.).

## Author contributions

M.A.M., M.N.S. and A.W. designed, performed and analysed the experiments with the help of B.A.T.W. and C.G.; T.H.H.C. performed computational analyses of sequencing data, with supervision of R.V-T. and help from B.A.T.W.; R.V-T. and C.S-S. helped with the collection of the samples for scRNAseq. M.N.S. and A.W. prepared illustrations. A.C., S.F., L.R., A.D., N.S., S.E., K.E., L.C., P.S. oversaw and provided human embryos for these studies. S.B. and M.Z-G. conceived, supervised the project and provided funding. The manuscript was assembled with the help of all authors.

## Competing interests

The authors declare no competing interests.
