## [Peer Review File · Nature Communications]

Reviewers' Comments:

Reviewer #2:

Remarks to the Author:

We were very pleased to see the authors response to our comments and are very happy for the manuscript to be published in Nature Communications.

Reviewer #3:

Remarks to the Author:

In this manuscript Mole et al. describe single-cell RNA sequencing analyses of a relatively large number of human embryos cultured in vitro into postimplantation stages (7, 9, 11dpf), as well as functional analyses of the role of FGF signaling in development of the three cellular lineages.

The significance of this work is at many levels. First, and given the scarcity of the human embryo specimens, the heterogeneity of in vitro development, and very limited studies carried out until now (Xiang et al., 2020 and Zhou et al., 2019; earlier work from these authors and Brivalou limited to antibody stainings), this work confirms and extends the knowledge on the changes in the pluripotency, as well as transcriptomes of the epiblast, hypoblast, and TE cell types at post-implantation stages. This work also highlights variability of in vitro cultured embryos, further underscoring the significance of such analyses.

Second, the authors reveal new roles of FGF signaling at these developmental stages that are distinct from those FGF plays in the mouse embryo: in human postimplantation embryos FGF promotes proliferation of embryonic and extraembryonic tissues.

Third, this work further characterizes the extra-embryonic hypoblast cell population, especially those cells that are in contact with epiblast, to show that these cells express/secrete inhibitors of BMP, NODAL and WNT signaling. The authors provide evidence in support of a biased distribution of these specialized hypoblast cells, leading to the proposal that a biased antagonist expression could underlie symmetry breaking in the human embryo preceding the process of gastrulation, akin to the function of the AVE in murine embryos.

Having reviewed both the manuscript and the response of the authors to the previous reviews, the manuscript is significantly improved by additional data, including additional embryo specimens, and additional analyses. The manuscript should be of significant interest to the fields of developmental and reproductive biology with implications for human reproduction. The manuscript is suitable for publication when the following few points are addressed.

In agreement with reviewers, the authors modified and toned down some of their conclusions. However, some conclusions in the abstract and main text still go beyond the experimental evidence provided. In particular, the statement "we describe the sequence of events that lead to symmetry breaking in the human embryo prior to gastrulation" and "a subset of extra-embryonic hypoblast cells, the anterior hypoblast, secretes inhibitors of the BMP, NODAL and WNT signalling pathways in a localised manner, leading to symmetry breaking and the initiation of epiblast patterning", should be expressed more cautiously". The main reason for tempering these

conclusions are the data presented on the biased expression of CER expressing cells. From 28 analyzed 9 dpf embryos, 10 showed distribution significantly different from 90 degrees but 17 were ns from 90 degrees, and 1 embryo could not be analyzed statistically. While this shows a trend when considering the symmetric distribution documented for 7dpf embryos, it does not provide definitive evidence. The sentence "These CER1 positive cells showed a localisation bias towards one side of the hypoblast (Fig. 3h, Supplementary Fig. 7d-f)" could be rephrased to report the results more precisely: "These CER1 positive cells showed a localization bias towards one side of the hypoblast in a subset of embryos (10/27)."

Likewise, the authors should reconsider the use of "anterior hypoblast" in the title, subheading "Molecular characterization of the anterior hypoblast signaling centre", and other uses of "anterior hypoblast signaling centre". "putative anterior signaling centre" as used in other instances is more consistent with the data. As 9 or 11dpf embryos do not show anteroposterior polarity, "anterior signaling centre" is a supposition based on mouse and monkey studies.

Moreover, isn't the conclusion "We confirmed that cells secreting LEFTY protein at 9 d.p.f. also displayed a localisation bias towards one side of the hypoblast in a similar way to CER1 secreting cells (Fig. 3j, k).", based on a staining of a single embryo too firm?

Tempering these conclusions would not take away from the significance and value of this work, which provides important insights into these mysterious stages of human development.

Reviewer #5:

Remarks to the Author:

Drawing from the results of single-cell transcriptome of the in vitro cultured embryos, this study showed the presence of the four major cell types that have been identified in 9 and 11 dpf stage human embryo. The results also reiterated the previous findings of the transition of pluripotency state during post-blastocyst development and the presence of a sub-population of hypoblast that displays the properties of anterior visceral endoderm of other model organisms. Experimental studies further showed that FGF signalling activity may play a role in the expansion and maintenance of the epiblast and the hypoblast in the post-implantation human embryo.

Analysis of the expression pattern of pluripotency markers of the epiblast of embryos equivalent to 9 and 11 dpf corroborated the prior knowledge of the transition of pluripotency state reported for human embryos developing in vitro and in vivo

While the transition affirmed the alignment of the differentiating epiblast to the expected developmental trajectory, it is unclear how far the transition from naïve to primed transition has proceeded between these two timepoints. The regression analysis showed that conventional (H9) hESC and reset ESCs shared transcriptome features with cells of all four embryonic stages, albeit both stem cell types may have a stronger correlation with the 6-7 dpf epiblast (Fig 1g, Suppl fig 5a, also for other hESCs). These results did not support the claim: "Conventional primed human embryonic stem cells (ESCs) shared transcriptional similarities with the post-implantation human epiblast at 11 d.p.f., while recently generated naïve human ESCs resembled the pre-implantation epiblast at 6-7 d.p.f. (Fig. 1g)." The inclusion of the 5 dpf ICM in the analysis added little to this study, which took the 6-7 dpf epiblast as the reference of the pluripotency state. It is likely that at 9-11 dpf, the epiblast is still in the process of transition, or that only sub-sets of epiblast cells has made the transition to the primed state. In view of that putative human formative pluripotent stem cells (XPSCs) has been reported, (Yu et al., 2021 doi.org/10.1016/j.stem.2020.11.003), this may offer an opportunity to benchmark the 9-11 dpf epiblast with the XPSCs, which may reveal if a formative cell state can be identified in the epiblast of the human post-blastocyst embryo.

The experiments using chemical inhibitors and FGF2/4 supplementation has informed the global requirement of FGF signalling in the expansion of the epiblast and hypoblast cell population.

Neither study has addressed the role of epiblast-specific FGF2/4 activity as (i) other FGF ligands are also expressed, although not as transcriptionally active as FGF2/4, and (ii) there are multiple FGFRs that may mediate the signalling activity of other ligands. The finding of "reduced variability in epiblast cell number" in the cultured embryos following supplementation of FGF2/FGF4 is not informative of the role of FGF signalling.

The emergence of a subset of CER1+/GATA6+ cells among the GATA6+ve hypoblast in 7 dpf embryo and the spatially biased localization of the CER1+ cells in 9 dpf and 11 dpf embryo are novel findings. Based on the transcriptome, these cells displayed the markers of mouse AVE, and expressed the genes encoding signalling antagonists known for the AVE. It is reasonable to presume that the secreted molecules are produced and may act on the epiblast cells in their proximity, as revealed by the changes in the signalling response genes. However, regarding the claim of "heterogeneity across the epiblast and spatially restricted expression of ID2", or ID3, ID4, SMAD3, SMAD1 in the same context: How could the heterogeneity of expression pattern of signalling targets among cells in the epiblast cluster (Suppl fig. 10) be correlated with spatially restricted response as would be predicted by prior knowledge of AVE function in other model organisms? In Fig 3I: Was CER1 expressed also in the OCT4+ cells, besides the GATA6+ve cells? Would the expression of pSMAD1.5 in the OCT4+ve cells in the proximity of the CER1+/GATA6+ cells point to the lack of response of the epiblast to the antagonistic activity?

Results on the emergence the AVE-like cells at an earlier timepoint added knowledge on the developmental scenario of the putative anterior hypoblast. However, it provided little insight into the "mechanism of formation of the human anterior hypoblast over time". Such information would have to be gleaned from the analysis of the regulons, and if feasible, functional characterization of the molecular pathway.

Specific points:

Were SFRP2 and SALL2 expression significantly increased (Fig. 1e)?

Was the changes in Dnmt3B expression consistent with data in Suppl Fig 4b?

The changes in "capacitated" cells (Suppl Fig 5b) has no direct relevance to the transition of pluripotency across two developmental ages/stages (day 9 and day 11) in this study.

".. particular enrichment of FGFR1 within the hypoblast": Was the enrichment of FGFR1 and FGFR2 in the hypoblast different?

It may be questioned if FGF18 expressed in the syncytiotrophoblast (Suppl Fig 6d) may play any role in post-blastocyst development.

Any quantified data supporting the claim of LEFTY1+ cells "displayed a localization bias"? (Fig. 3j, k)?

Pooled data on the distribution of CER1+ cells in 9 dpf embryo (Fig 3h): The data from embryos (n=10) showing significant bias from the 90o sector should be pooled separately from those of ns and N/A embryos (n=18) (Suppl Fig 7f). This may help highlighting that not all 9 dpf embryos were showing biased localization.

The mechanistic impact of a decrease in the proportion (and number) of CER1+ cells in the GATA6+ve population from 7 to 9dpf (69-82%: median cell number =14, to 17-43%: median cell number = 5) on the spatial distribution of the AVE-like cells is unknown. Might this suggest that the "asymmetrical / biased" distribution of the CER1+ve cells is the consequence of uneven expansion of the GATA6+/CER1-ve cells and/or loss of CER1+ve cells?

Cell type identity:

- Were amnion epithelial cells present in 11 dpf embryos in vitro?
- If some of the sub-cluster 0 hypoblast cells are the AVE-like cells or the anterior hypoblast precursors (Fig. 3a-d, Suppl Fig 7a-c), what may the rest of sub-cluster 0 and sub-clusters 1 and 2 hypoblast be in the d9/d11 pdf embryo (Suppl Fig 9a)? Do they display different regulons from the CER1+ cells?

Descriptors/annotations that may not be appropriate/correct:

- "free-floating" blastocysts – are they free-floating?
- Fig. 1b, "bold red": only "bold" was indicated in the legend; Did "red" color refer to gene name or color legend? Less than top ten genes (red?) were identified for cytotrophoblast and hypoblast; CDH1 was not among the cytotrophoblast enriched markers.
- Suppl Fig. 7f legend: clarify "against the proximal plane (90o)" – Was it the 90o sector relative to the proximal plane?
- "... while the opposing region becomes the site of the primitive streak formation": might "opposing region" be "region of the epiblast distant from the AVE becomes the prospective posterior side of the embryo"?
- Suppl Fig 11: Would the CER1+ cells in 7 dpf embryos be warranted to be "anterior hypoblast" if they are not biased in distribution (Suppl Fig. 12c)?
- Text: Missing reference to Suppl Fig 1f, g
- Were Bars in Fig 1d, e, f = median?

Responses to Reviewers' comments - NCOMMS-20-49409-T

Previous title of the manuscript: Identification of the Anterior Hypoblast and Signalling Interactions Driving Morphogenesis of the Human Embryo

New proposed title: Single Cell Analysis of Human Embryos Defines the Putative Anterior Hypoblast Signalling Centre

Date: 15-03-2021

Please note that Reviewers' comments are in plain black text, while our responses are in blue text.

Modifications introduced in the main text of the manuscript are highlighted in red.

Reviewer #2 (Remarks to the Author):

We were very pleased to see the authors response to our comments and are very happy for the manuscript to be published in Nature Communications.

We thank the Reviewer for the very useful comments during the revisions which enable us to improve the manuscript and introduce important experiments.

Reviewer #3 (Remarks to the Author):

In this manuscript Mole et al. describe single-cell RNA sequencing analyses of a relatively large number of human embryos cultured in vitro into postimplantation stages (7, 9, 11dpf), as well as functional analyses of the role of FGF signaling in development of the three cellular lineages.

The significance of this work is at many levels. First, and given the scarcity of the human embryo specimens, the heterogeneity of in vitro development, and very limited studies carried out until now (Xiang et al., 2020 and Zhou et al., 2019; earlier work from these authors and Brivnalou limited to antibody stainings), this work confirms and extends the knowledge on the changes in the pluripotency, as well as transcriptomes of the epiblast, hypoblast, and TE cell types at post-implantation stages. This work also highlights variability of in vitro cultured embryos, further underscoring the significance of such analyses.

Second, the authors reveal new roles of FGF signaling at these developmental stages that are distinct from those FGF plays in the mouse embryo: in human postimplantation embryos FGF promotes proliferation of embryonic and extraembryonic tissues.

Third, this work further characterizes the extra-embryonic hypoblast cell population, especially those cells that are in contact with epiblast, to show that these cells express/secrete inhibitors of BMP, NODAL and WNT signaling. The authors provide evidence in support of a biased distribution of these specialized hypoblast cells, leading to the proposal that a biased antagonist expression could underlie symmetry breaking in the human embryo preceding the process of gastrulation, akin to the function of the AVE in murine embryos.

Having reviewed both the manuscript and the response of the authors to the previous reviews, the manuscript is significantly improved by additional data, including additional embryo specimens, and additional analyses. The manuscript should be of significant interest to the fields of developmental and reproductive biology with implications for human reproduction. The manuscript is suitable for publication when the following few points are addressed.

In agreement with reviewers, the authors modified and toned down some of their conclusions. However, some conclusions in the abstract and main text still go beyond the experimental evidence provided. In particular, the statement “we describe the sequence of events that lead to symmetry breaking in the human embryo prior to gastrulation” and “a subset of extra-embryonic hypoblast cells, the anterior hypoblast, secretes inhibitors of the BMP, NODAL and WNT signalling pathways in a localised manner, leading to symmetry breaking and the initiation of epiblast patterning”, should be expressed more cautiously”. The main reason for tempering these conclusions are the data presented on the biased expression of CER expressing cells. From 28 analyzed 9 dpf embryos, 10 showed distribution significantly different from 90 degrees but 17 were ns from 90 degrees, and 1 embryo could not be analyzed statistically. While this shows a trend when considering the symmetric distribution documented for 7dpf embryos, it does not provide definitive evidence. The sentence “These CER1 positive cells showed a localisation bias towards one side of the hypoblast (Fig. 3h, Supplementary Fig. 7d-f)” could be rephrased to report the results more precisely: “These CER1 positive cells showed a localization bias towards one side of the hypoblast in a subset of embryos (10/27).”

We thank the Reviewer for the feedback, and we have modified the text as suggested. Below are the modifications introduced. They are highlighted in red in the manuscript for tracking.

Abstract:

<<Here, through single-cell sequencing of human embryos developing ex vivo, coupled with functional characterisation studies, we describe key events of human embryo morphogenesis in the period between implantation and gastrulation.>>

<<In a subset of embryos, we identified a group of extra-embryonic hypoblast cells that expresses inhibitors of the BMP, NODAL and WNT signalling pathways in a localised manner. This putative anterior hypoblast could act as a signalling centre for the initiation of epiblast patterning prior to gastrulation.>>

Text: <<These CER1 positive cells showed a localisation bias towards one side of the hypoblast in a subset of embryos.>>

Full details of the number and quantifications are reported in the figure legends:

Figure 3: << Number of embryos n=28, data combined from individual embryo quantification from Supplementary Figure 7f. CER1 positive cells showed a statistically significant localisation bias towards one side of the hypoblast in 10 out of 28 embryos analysed.>>

Suppl Figure 7f: <<Angular distribution of CER1 positive cells along the hypoblast hemisphere (0° to 180°) calculated for individual embryos at 9 d.p.f. and represented as summary combined data in Figure 3h. Biases in the angular distribution of CER1 cells against the distal plane (90°) for each embryo calculated by one sample T-test: 10 embryos are statistically different from 90° degree: MZG949: p=0.0075 **, MZG944: p=0.0059 **, MZG937: p=0.0068 **, MZG827: p<0.0001****, MZG968: p<0.0096**, MZG954: p<0.0001****, MZG418-1: p<0.005***, MZG448: p<0.0112*, MZG411: p<0.0001****, MZG502: p<0.0001****; 17 embryos are ns different from 90° degree, p=ns in MZG942, MZG941, MZG939, MZG412, MZG738, MZG816, MZG824-a, MZG418-2, MZG953, MZG967 MZG966, MZG955, MZG 591, MZG952, MZG895, MZG898, MZG852; MZG899: the n number is too small and could not be analysed statistically.>>

Likewise, the authors should reconsider the use of “anterior hypoblast” in the title, subheading “Molecular characterization of the anterior hypoblast signaling centre”, and other uses of “anterior hypoblast signaling centre” . “putative anterior signaling centre” as used in other instances is more consistent with the data. As 9 or 11dpf embryos do not show anteroposterior polarity, “anterior signaling centre” is a supposition based on mouse and monkey studies.

We have corrected the nomenclature as “putative anterior hypoblast signalling centre” to better reflect the data of the manuscript, as suggested by the Reviewer.

The title has been modified as: << Single Cell Analysis of Human Embryos Defines the Putative Anterior Hypoblast Signalling Centre >>

Subheading: <<Molecular characterisation of the putative anterior hypoblast signalling centre>>

These modifications have been introduced throughout the entire text of the manuscript.

Moreover, isn't the conclusion “We confirmed that cells secreting LEFTY protein at 9 d.p.f. also displayed a localisation bias towards one side of the hypoblast in a similar way to CER1 secreting cells (Fig. 3j, k).”, based on a staining of a single embryo too firm?

We thank the Reviewer for this comment. We have carried out analysis of angle distribution for Lefty1 in the same manner as for Cer1 (no of embryos= 7. Details of angle distribution for each embryo is reported in Suppl Figure 10. The averaged

distribution among the 7 samples has been included in Figure 3I. Cells secreting Lefty exhibit a similar pattern to Cer1 expressing cells.

Tempering these conclusions would not take away from the significance and value of this work, which provides important insights into these mysterious stages of human development.

We thank the Reviewer for the feedback who helped us to improve the manuscript in several instances.

Reviewer #5 (Remarks to the Author):

Drawing from the results of single-cell transcriptome of the in vitro cultured embryos, this study showed the presence of the four major cell types that have been identified in 9 and 11 dpf stage human embryo. The results also reiterated the previous findings of the transition of pluripotency state during post-blastocyst development and the presence of a sub-population of hypoblast that displays the properties of anterior visceral endoderm of other model organisms. Experimental studies further showed that FGF signalling activity may play a role in the expansion and maintenance of the epiblast and the hypoblast in the post-implantation human embryo.

Analysis of the expression pattern of pluripotency markers of the epiblast of embryos equivalent to 9 and 11 dpf corroborated the prior knowledge of the transition of pluripotency state reported for human embryos developing in vitro and in vivo. While the transition affirmed the alignment of the differentiating epiblast to the expected developmental trajectory, it is unclear how far the transition from naïve to primed transition has proceeded between these two timepoints. The regression analysis showed that conventional (H9) hESC and reset ESCs shared transcriptome features with cells of all four embryonic stages, albeit both stem cell types may have a stronger correlation with the 6-7 dpf epiblast (Fig 1g, Suppl fig 5a, also for other hESCs). These results did not support the claim: “Conventional primed human embryonic stem cells (ESCs) shared transcriptional similarities with the post-implantation human epiblast at 11 d.p.f., while recently generated naïve human ESCs resembled the pre-implantation epiblast at 6-7 d.p.f. (Fig. 1g).” The inclusion of the 5 dpf ICM in the analysis added little to this study, which took the 6-7 dpf epiblast as the reference of the pluripotency state. It is likely that at 9-11 dpf, the epiblast is still in the process of transition, or that only sub-sets of epiblast cells has made the transition to the primed state. In view of that putative human formative pluripotent stem cells (XPSCs) has been reported, (Yu et al., 2021 doi.org/10.1016/j.stem.2020.11.003), this may offer an opportunity to benchmark the 9-11 dpf epiblast with the XPSCs, which may reveal if a formative cell state can be identified in the epiblast of the human post-blastocyst embryo.

We thank the Reviewers for the feedback and we have modified the statement as follows: << Despite similarities with both pre and post-implantation stages (5, 6-7, 9 and 11 d.p.f), conventional primed human embryonic stem cells (ESCs) shared transcriptional similarities with the post-implantation human epiblast at 11 d.p.f., while recently generated naïve human ESCs resembled the pre-implantation epiblast at 6-7 d.p.f. more closely (Fig. 1g).>> We have added transcriptional similarity analyses

including all the stages as suggested by the Reviewer. The regression analysis in Figure 1g supports the correlation of conventional primed cells with 11 d.p.f while the naïve human ESCs (Rset) correlate with 6-7 d.p.f. The difference between the two cell types is statistically significant at these two stages.

Below, we provide new regression analysis, as suggested by the Reviewer, comparing our embryo dataset with the new XPSCs line, purportedly representing the intermediate formative stage between naïve and primed pluripotency (Yu et al 2021). There is a strong correlation of these cells with the late post-implantation epiblast at day 11. However, it is surprising that XPSCs line share more similarities with pre-implantation epiblast at day 6-7, rather than the epiblast day 9. In the absence of data from matched control hiPSCs cultured in primed conditions (which is not provided in Yu et al 2021) it is difficult to draw conclusions from this analysis. Moreover, the *in vivo* relevance of the formative state in humans is still under debate and therefore, we have decided not to include this analysis in the manuscript.

References:

L. Yu et al., Derivation of Intermediate Pluripotent Stem Cells Amenable to Primordial Germ Cell Specification. *Cell Stem Cell* (2020), doi:10.1016/j.stem.2020.11.003.

The experiments using chemical inhibitors and FGF2/4 supplementation has informed the global requirement of FGF signalling in the expansion of the epiblast and hypoblast cell population. Neither study has addressed the role of epiblast-specific FGF2/4 activity as (i) other FGF ligands are also expressed, although not as transcriptionally active as FGF2/4, and (ii) there are multiple FGFRs that may mediate the signalling activity of other ligands. The finding of “reduced variability in epiblast cell number” in the cultured embryos following supplementation of FGF2/FGF4 is not informative of the role of FGF signalling.

We acknowledge in the text that there are other FGF ligands expressed and provide these data in Suppl Figure 6a-e <<The epiblast preferentially expressed low levels of additional FGF ligands FGF8, FGF13, FGF17, and FGF19 (Supplementary Fig. 6a-e).>> We agree with the Reviewer that there are multiple FGFRs as shown in Figure 2b. Because of this heterogeneity we used the pan-FGFR inhibitor, LY28744549, to target all the four receptors. We hope that future studies will be able to address more specifically the role of each FGF receptor and ligand.

The emergence of a subset of CER1+/GATA6+ cells among the GATA6+ve hypoblast in 7 dpf embryo and the spatially biased localization of the CER1+ cells in 9 dpf and 11 dpf embryo are novel findings. Based on the transcriptome, these cells displayed the markers of mouse AVE, and expressed the genes encoding signalling antagonists known for the AVE. It is reasonable to presume that the secreted molecules are produced and may act on the epiblast cells in their proximity, as revealed by the changes in the signalling response genes. However, regarding the claim of “heterogeneity across the epiblast and spatially restricted expression of ID2”, or ID3, ID4, SMAD3, SMAD1 in the same context: How could the heterogeneity of expression pattern of signalling targets among cells in the epiblast cluster (Suppl fig. 10) be correlated with spatially restricted response as would be predicted by prior knowledge of AVE function in other model organisms?

This analysis is based on embryos which were single cell sequenced. As these were dissociated in single cell, we lose the spatial reference of the epiblast cells in proximity to the putative anterior hypoblast vs the ones far away. This is a limitation. However, the spatially restricted in this case refers to the cells within the cluster. As this may generate confusion, we have corrected the sentence as << we observed heterogeneity across the epiblast and restricted expression of ID2 in a subset of cells>>

In Fig 3l: Was CER1 expressed also in the OCT4+ cells, besides the GATA6+ve cells?

This is a maximum projection, and this is why it may give the appearance of CER1 expression on the OCT4 cells. CER1 is only associated with the GATA6+ cells. Below is an example of single slice through the Z stack.

We have specified in the picture that this is a maximum projection.

Red: OCT4, Grey: CER1, Green: pSMAD1.5

Would the expression of pSMAD1.5 in the OCT4+ve cells in the proximity of the CER1+/GATA6+ cells point to the lack of response of the epiblast to the antagonistic activity?

pSMAD1.5+ is majorly expressed in OCT4 cells distant from the CER1 site (mid-right part of Figure 3m), not in proximity of CER1+ cells. As CER1 is secreted by the anterior hypoblast centre, is expected to create a morphogen gradient through the epiblast. Due to the use of max projections, it may be difficult to distinguish the expression of

pSMAD1.5. As such we provide a series of single slices in Suppl Figure 10b. We have also provided a further example (embryo ID: MZG 895).

Results on the emergence the AVE-like cells at an earlier timepoint added knowledge on the developmental scenario of the putative anterior hypoblast. However, it provided little insight into the “mechanism of formation of the human anterior hypoblast over time”. Such information would have to be gleaned from the analysis of the regulons, and if feasible, functional characterization of the molecular pathway.

We agree that more work would be needed in the future to investigate the formation of the human anterior hypoblast over time. We also point out in the discussion that it would be important in future studies to determine whether the shift in localisation towards one side of the epiblast at 9 d.p.f occurs as a result of active migration as in the mouse, or due to local downregulation of CER1 from an initially widespread expression domain, causing only one side of the hypoblast to retain its expression. We agree that our manuscript is not providing functional mechanistic data regarding the emergence of the putative anterior hypoblast, and therefore we have removed this claim from the manuscript.

Specific points:

Were SFRP2 and SAIL2 expression significantly increased (Fig. 1e)?

Yes. We have checked by performing non parametric one-way ANOVA (Kruskall Wallis Test) as below, confirming a difference of day 5 and 6-7 with day 9 and 11.

For SFRP2

Dunn's multiple comparisons test	Mean rank	Significant?	Summary	Adjusted P Value
5 d.p.f vs. 6-7 d.p.f	-32.82	No	ns	0.1218
5 d.p.f vs. 9 d.p.f	-90.11	Yes	****	<0.0001
5 d.p.f vs. 11 d.p.f	-70.06	Yes	****	<0.0001
6-7 d.p.f vs. 9 d.p.f	-57.29	Yes	****	<0.0001
6-7 d.p.f vs. 11 d.p.f	-37.24	Yes	*	0.0257
9 d.p.f vs. 11 d.p.f	20.05	No	ns	0.4464

For SAIL2

5 d.p.f vs. 6-7 d.p.f	-13.16	No	ns	>0.9999
5 d.p.f vs. 9 d.p.f	-59.81	Yes	****	<0.0001
5 d.p.f vs. 11 d.p.f	-41.56	Yes	**	0.0041
6-7 d.p.f vs. 9 d.p.f	-46.65	Yes	****	<0.0001
6-7 d.p.f vs. 11 d.p.f	-28.41	No	ns	0.0804
9 d.p.f vs. 11 d.p.f	18.24	No	ns	0.3936

All the raw values are provided in Supplementary Table 4.

Was the changes in Dnmt3B expression consistent with data in Suppl Fig 4b?

The data presented in Suppl Figure 4b refers to the expression of naïve and primed pluripotency markers in Xiang et al. 2020 dataset, not our dataset reported in this manuscript. Both our dataset and the Xiang et al., 2020 datasets seem to be quite consistent as Dnmt3B is less expressed in ICM and pre-Epi as opposed to post-Epi. We do not have a comparative stage in Figure 1e which corresponds to the late primitive streak epiblast stage shown in Suppl Figure 4b. However, we also see a decrease at later stages (day 11).

The changes in “capacitated” cells (Suppl Fig 5b) has no direct relevance to the transition of pluripotency across two developmental ages/stages (day 9 and day 11) in this study.

Upon capacitation, cells lose their naïve pluripotency character and transition into post-implantation-like stages. The logistic regression analysis shows that capacitated cells acquire a post-implantation-like day 11 pluripotent gene signature (Suppl Fig 5b). We have clarified this in the text. This is consistent with what is observed in supplementary figure 5a: primed cells show close similarity to day 11 only.

“.. particular enrichment of FGFR1 within the hypoblast”: Was the enrichment of FGFR1 and FGFR2 in the hypoblast different?

Yes, the expression levels differ. FGFR1 is more strongly expressed in the hypoblast. FGFR2 has a lower expression level and more heterogenous expression. We provide these data in Supplementary Table 5.

Scaled Averages					Raw Expression Averages (log2 Integrated counts)				
FGFRL1	0.032496	-0.10496	-0.25228	0.172066	FGFRL1	0.114745	0.060523	0.03162	0.148196
FGFR1	-0.30274	0.550218	1.279475	2.608788	FGFR1	0.355312	0.892915	1.84297	4.768168
FGFR2	-0.20286	0.245037	0.97065	2.444652	FGFR2	0.037897	0.112538	0.287342	0.719355
FGFR3	-0.06011	-0.29565	1.439043	1.840475	FGFR3	0.079044	0.013541	0.418327	0.562894
FGFR4	-0.18879	0.257141	1.167391	1.759851	FGFR4	0.101696	0.18851	0.482219	0.733702

It may be questioned if FGF18 expressed in the syncytiotrophoblast (Suppl Fig 6d) may play any role in post-blastocyst development.

Yes, it is a very interesting observation. Future studies to block this specific FGF ligand will be needed to address this point.

Any quantified data supporting the claim of LEFTY1+ cells “displayed a localization bias”? (Fig. 3j, k)?

We have carried out an angle analysis similar to what has been done for Cer1 expression and confirm that Lefty1 expressing cells displayed a localisation bias as Cer1. Details of angle distribution for each embryo is reported in Suppl Figure 10a. The averaged distribution among the 7 samples has been included in figure 3l. Cells secreting Lefty exhibit a similar pattern to Cer1 expressing cells.

Pooled data on the distribution of CER1+ cells in 9 dpf embryo (Fig 3h): The data from embryos (n=10) showing significant bias from the 90o sector should be pooled separately from those of ns and N/A embryos (n=18) (Suppl Fig 7f). This may help highlighting that not all 9 dpf embryos were showing biased localization.

As we provide the quantification for each single embryo in Suppl Fig 7f, and we have emphasised this heterogeneity in the text and the figure legend we believe it will be appropriate to have an overall pooled graph as shown in Figure 3h, regardless of the statistical significance of each embryo, as this will represent a summary of all embryos analysed rather than a selection of a group of embryos.

The mechanistic impact of a decrease in the proportion (and number) of CER1+ cells in the GATA6+ve population from 7 to 9dpf (69-82%: median cell number =14, to 17-43%: median cell number = 5) on the spatial distribution of the AVE-like cells is unknown. Might this suggest that the “asymmetrical / biased” distribution of the CER1+ve cells is the consequence of uneven expansion of the GATA6+/CER1-ve cells and/or loss of CER1+ve cells?

Yes, that is a possibility that some cells will retain CER1 expression while others will suppress it, causing downregulation of this gene on one side. We now discuss this possibility.

Cell type identity:

- Were amnion epithelial cells present in 11 dpf embryos in vitro?

No, we do not see amnion cells. Currently there are very few definitive markers which define the amnion, and these studies are primarily based on morphological observations (squamous amnion cells vs columnar epithelium of the epiblast disk). One promising marker is ISL1, which is reported in both primates and human stem cell derived models (Yang et al., 2021). We do not see strong enrichment of ISL1 in any subset of cells in our dataset. Also, we have not addressed this point in this study as opposed to Xiang et al 2020. Recently, it has been reported that Xiang et al., 2020 works might have misclassified their purported amnion cells as these were largely syncytiotrophoblast cells (Chhabra and Warmflash, 2021). We have been careful as to not make these types of claims regarding the presence of more rare cell types in our single cell sequencing data.

References:

- Chhabra, S., and Warmflash, A. (2021). BMP-treated human embryonic stem cells transcriptionally resemble amnion cells in the monkey embryo. *BioRxiv* 2021.01.21.427650.
- Deglincerti, A., Croft, G.F., Pietila, L.N., Zernicka-Goetz, M., Siggia, E.D., and Brivanlou, A.H. (2016). Self-organization of the in vitro attached human embryo. *Nature* 533, 251.
- Yang, R., Goedel, A., Kang, Y., Si, C., Chu, C., Zheng, Y., Chen, Z., Gruber, P.J., Xiao, Y., Zhou, C., et al. (2021). Amnion signals are essential for mesoderm formation in primates. *BioRxiv* 2020.05.28.118703.

- If some of the sub-cluster 0 hypoblast cells are the AVE-like cells or the anterior hypoblast precursors (Fig. 3a-d, Suppl Fig 7a-c), what may the rest of sub-cluster 0 and sub-clusters 1 and 2 hypoblast be in the d9/d11 pdf embryo (Suppl Fig 9a)? Do they display different regulons from the CER1+ cells?

We have not characterised the identity of subcluster 1 and 2 in this manuscript. However, we have performed differential expression analysis between each of the subclusters of the hypoblast as provided in Supplemental Table 8. For subcluster 1 this includes *NID2*, *LAMC1*, *LAMA1*. For subcluster 2 this includes *CGA*, *PGF*, *CEBPB*.

Further, below we show the expression of several genes that are differentially expressed across the hypoblast subclusters. As shown below, BMP signaling target *ID3* and Wnt receptor *FRZB* are upregulated in subcluster 1. Interestingly, trophoblast markers *GATA2*, *KRT18*, and *GATA3* are enriched in subcluster 2.

Regarding regulons, based on roc analysis, the top 3 most differentially active regulons in subcluster 1 is: SMAD3, PITX2, CDX2; In subcluster 2: GABPB1, RFXANK, THAP11.

Given the enriched SMAD3 activity and *ID3* expression, we can hypothesize that the cells in subcluster 1 have increased BMP signalling activity, and perhaps represent the posterior hypoblast. Further, given the enrichment of trophoblast markers in subcluster 2, these cells may represent the unique “yolk sac trophoblast” (ysTE) cell type previously described in *ex vivo* cultured human embryos (Deglincerti et al., 2016). Given that we have not characterised markers of these subclusters at a protein level/spatially, we have not speculated about their identity in the manuscript.

Descriptors/annotations that may not be appropriate/correct:

- “free-floating” blastocysts – are they free-floating?

Yes, we use this terminology to describe the fact that they have not attached yet. This can be observed at blastocyst stage day 5 and 6.

- Fig. 1b, “bold red”: only “bold” was indicated in the legend; Did “red” color refer to gene name or color legend? Less than top ten genes (red?) were identified for cytotrophoblast and hypoblast; CDH1 was not among the cytotrophoblast enriched markers.

We have corrected this with bold red. It refers to gene name.

We have also eliminated the word top ten genes, and refers only to top most enriched genes.

Regarding *CDH1* expression in the cytotrophoblast, this is due to the inclusion of all clusters in the differential expression analysis. As visible below, *CDH1* is expressed strongly in the cytotrophoblast, epiblast, and to a slightly lesser extent in the hypoblast. *CDH1* is considered a cytotrophoblast marker relative to other trophoblast derivative populations – not compared to the epiblast or hypoblast. Indeed, when performing a pairwise statistical test between cytotrophoblast and syncytiotrophoblast, *CDH1* comes up as a positive marker for cytotrophoblast.

- Suppl Fig. 7f legend: clarify ‘ against the proximal plane (90o)’ – Was it the 90o sector relative to the proximal plane?

We have corrected as 90 degrees refers to the distal plane.

- “.. while the opposing region becomes the site of the primitive streak formation”: might “opposing region” be “region of the epiblast distant from the AVE becomes the prospective posterior side of the embryo”?

Yes we have corrected as << while the distal region becomes the site of the primitive streak formation>>

- Suppl Fig 11: Would the CER1+ cells in 7 dpf embryos be warranted to be “anterior hypoblast” if they are not biased in distribution (Suppl Fig. 12c)?

We agree with the Reviewer and corrected the title as << The hypoblast signalling centre>>.

- Text: Missing reference to Suppl Fig 1f, g

We have introduced in the text: <<and were therefore excluded from downstream analyses (Supplementary Fig. 1a-g).>>

- Were Bars in Fig 1d, e, f = median?

Yes they are medians, we apologies and have introduced this in the figure legend.

Reviewers' Comments:

Reviewer #3:

Remarks to the Author:

In this revised manuscript the authors made a strong effort to address the reviewers' concerns. The manuscript provides additional highly valuable datasets of single-cell transcriptomes of in vitro cultured post implantation human embryos at 9 and 11dpf. The authors also provide evidence that inhibition of signaling by FGF ligands expressed by epiblast limits cell proliferation in all lineages at 6-8dpfs. In addition the authors provide several lines of evidence in support of a subset of hypoblast cells expressing CER1 and other inhibitors of BMP, NODAL and WNT that is broadly distributed at 7dpf but becomes asymmetric by 9dpf. Providing experimental support for the concept of a putative anterior hypoblast signaling center is an important advance in understanding human development.

The manuscript is suitable for publication, but addressing the following textual or figure annotation points would further improve the manuscript and its readability.

In the revised manuscript the authors note "The epiblast preferentially expressed low levels of additional FGF ligands FGF8, FGF13, FGF17, and FGF19 (Supplementary Fig. 6a-e). The authors need to reconsider this interpretation of FGF13 as a FGF ligand, as FGF13 belongs to the class of intracellular FGF, iFGFs, which are not secreted and have no identified interaction with signaling FGFRs. FGF13 is known to bind microtubules and is involved in both their polymerization and stabilization (Ornitz DM, Wires Dev Bio, 2015).

The inhibition of FGF signaling experiments lead the authors to suggest that "FGF signaling is necessary for proliferation of epiblast, hypoblast and trophoblast lineages". It is interesting then that excess of FGF2 and FGF4 ligands does not lead to increased cell number in these lineages, and it seems to reduce the number of cells in the epiblast? (Sup. Figure 6h). How these results can be reconciled should be discussed.

The Results state "the mouse AVE marker HHEX, as well as FOXA3, LHX1, MIXL1, MNX1, and SOX21 did show preferential activity in the putative anterior hypoblast sub-cluster (sub-cluster 0)(Supplementary Fig. 8h-m). However, the subcluster 0 is not annotated in Supplemental Figure 8.

Similarly, Supplementary Figure 9a shows a heatmap with "the most enriched genes that characterize the sub-cluster 0", but in Supplementary Figure 9b-d UMAPs, the subcluster 0 (or other subclusters shown in S Figure 9a, are not annotated.

The following sentence in the last paragraph of Discussion needs rephrasing "While our data highlights the possible function of the hypoblast in mediating epiblast patterning via secreted inhibitors, while the role of the trophoblast and amnion as a source of BMP and WNT signals remains to be explored."

Reviewer #5:

Remarks to the Author:

The manuscript was revised to address the issues raised in previous review, including the concern of over-reaching data interpretation and inferences.

Points for further consideration:

It may be questioned whether the findings of this study are relevant to the delineation of the

“signalling centre” (page 9) in post-implantation human embryos, beyond the characterization of the anterior hypoblast-like cells (re: sub-cluster 0, Suppl Fig 9, which as referred to as the “putative human anterior hypoblast” – page 9 paragraph 1, page 11 paragraph 2). The presumption that that the regionalized cells act as a signalling source is based on the expression profile of signal-response factors in the epiblast.

As the asymmetrical distribution of the anterior hypoblast-like cells was observed in “a subset of embryos” can be gleaned from the dataset shown in Suppl Fig 7. The pooled data (Fig. 3h) of a selected set of embryos pre-judged to have displayed such asymmetry did not add more to the inference of biased distribution of these hypoblast cells in some embryos. Regarding the distribution of LEFTY-positive cells, which subset of embryos (only 2 among those in Suppl Fig 10?) showed the biased distribution? Were these embryos also displayed biased distribution of “CERL1-secreting” cells, in view of the single-cell result of co-expression of CERL1 and LEFTY1/2?

Benchmarking the pluripotency state (Page 5-6): The characterization of cell state by similarities of transcriptome (“strongly resembled”) is equivocal. In essence, the epiblast cells of the 9 dpf and 11 dpf embryos are overall similar to H9 cells (and other conventional hESCs – Suppl Fig 5a), and they are also “significantly similar” (by correlation analysis) to H9 reset cells. The data did not support the inference of “indicate a progressive pluripotency transition” at 9 and 11 dpf. The epiblast cells in the cultured embryos are likely at an undefined (intermediate?) state (discussion: page 10 paragraph 1) that is shared in part with both H9 and H9 reset cells.

Qualify the descriptor of “preferentially” expressed low level of other FGFs – would “differentially” be appropriate? (Page 6)

What is the toxicity of “removing FGF signalling entirely” (Page 7)? Is dampening cell proliferation and thereby leading to reduced cell number a toxic effect? Supplementing exogenous FGF2 and FGF4 activity did not change cell number but it minimized the variation of epiblast cell number. Is a more consistent cell number expected of 9 and 11 dpf embryos in vivo (if such in vivo data are available)? Would the variation in epiblast and hypoblast cell number among the cultured embryos imply the culture has not provide optimal embryotrophic activity? How may this affect the identification of cell types, the relative abundance of specific cell type (if not all cells in the embryo have been sampled for analysis, Fig. 2d-f, Suppl Fig 6f-h), and the source and level of inter-cellular signalling activity?

Page 10-11: The results of the experiments of chemical inhibition and signalling factor supplementation did not warrant the inference of FGF signals are “emanating from the epiblast lineage” to regulate the development, in the context of proliferation or “fate specification” of three blastocyst cell lineages.

pSMAD1.5 results were shown in Fig. 3m, not 3l (page 9 paragraph 2)

Suppl Fig 5 c and d were missing (page 5 second last line).

Title:

A single cell characterisation of human embryogenesis identifies pluripotency transitions and putative anterior hypoblast centre

Date: 09-05-2021

Please note that Reviewers' comments are in plain black text, while our responses are in blue text.

Reviewer #3 (Remarks to the Author):

In this revised manuscript the authors made a strong effort to address the reviewers' concerns. The manuscript provides additional highly valuable datasets of single-cell transcriptomes of in vitro cultured post implantation human embryos at 9 and 11dpf. The authors also provide evidence that inhibition of signaling by FGF ligands expressed by epiblast limits cell proliferation in all lineages at 6-8dpfs. In addition the authors provide several lines of evidence in support of a subset of hypoblast cells expressing CER1 and other inhibitors of BMP, NODAL and WNT that is broadly distributed at 7dpf but becomes asymmetric by 9dpf. Providing experimental support for the concept of a putative anterior hypoblast signaling center is an important advance in understanding human development.

The manuscript is suitable for publication, but addressing the following textual or figure annotation points would further improve the manuscript and its readability.

In the revised manuscript the authors note "The epiblast preferentially expressed low levels of additional FGF ligands FGF8, FGF13, FGF17, and FGF19 (Supplementary Fig. 6a-e). The authors need to reconsider this interpretation of FGF13 as a FGF ligand, as FGF13 belongs to the class of intracellular FGF, iFGFs, which are not secreted and have no identified interaction with signaling FGFRs. FGF13 is known to bind microtubules and is involved in both their polymerization and stabilization (Ornitz DM, Wires Dev Bio, 2015).

We thank the Reviewer for these supportive comments. We agree and indeed because FGF13 is not a secreted ligand but an intracellular microtubule-binding protein, as pointed out by the Reviewer, we have reconsidered our interpretation and removed FGF13 and focused only the secreted FGFs interacting with FGFRs.

The inhibition of FGF signaling experiments lead the authors to suggest that “FGF signaling is necessary for proliferation of epiblast, hypoblast and trophoblast lineages”. It is interesting then that excess of FGF2 and FGF4 ligands does not lead to increased cell number in these lineages, and it seems to reduce the number of cells in the epiblast? (Sup. Figure 6h). How these results can be reconciled should be discussed.

The data in Suppl Fig 6h (now Suppl Fig 6g) show a very mild decrease in the mean value of epiblast cells upon FGF2/4 treatment. While an F test for variance is significant between these groups, a two-tailed Welch’s T-test comparing the means is insignificant with a p-value of 0.31. We also supply all the raw data in the Supplementary Data 6. We find that the addition of FGF2/4 decreases the variability in cell numbers but not the average. This variability could be due to differences in embryo quality among patients and also the effect of the culture conditions, which we discuss in the methods section. We have seen a similar variability in terms of expression levels of FGF2 and FGF4 in the epiblast (please see Figure 2a, zoom in). This variability may in part explain the variability in cell numbers, which is reduced upon addition of exogenous FGF2.

The Results state “the mouse AVE marker HHEX, as well as FOXA3, LHX1, MIXL1, MNX1, and SOX21 did show preferential activity in the putative anterior hypoblast sub-cluster (sub-cluster 0)(Supplementary Fig. 8h-m). However, the subcluster 0 is not annotated in Supplemental Figure 8.

Similarly, Supplementary Figure 9a shows a heatmap with “the most enriched genes that characterize the sub-cluster 0”, but in Supplementary Figure 9b-d UMAPs, the subcluster 0 (or other subclusters shown in S Figure 9a, are not annotated.

We thank the Reviewer for pointing this out and apologise. The identity of the sub-clusters is provided in Figure 3a. We have clarified this point in the corresponding figure legends of Supplemental Figure 8 and 9, but not annotated within the figures themselves. Instead we refer to Figure 3a, which shows the identity of the sub-clusters.

The following sentence in the last paragraph of Discussion needs rephrasing “While our data highlights the possible function of the hypoblast in mediating epiblast patterning via secreted inhibitors, while the role of the trophoblast and amnion as a source of BMP and WNT signals remains to be explored.”

We thank the refers for this observation. We have rephrased and corrected the misspelling: <<While our data highlights the possible function of the hypoblast in mediating epiblast patterning via secreted inhibitors, the role of the trophoblast and amnion as a source of BMP and WNT signals remains to be explored>>

Reviewer #5 (Remarks to the Author):

The manuscript was revised to address the issues raised in previous review, including the concern of over-reaching data interpretation and inferences.

Points for further consideration:

It may be questioned whether the findings of this study are relevant to the delineation of the “signalling centre” (page 9) in post-implantation human embryos, beyond the characterization of the anterior hypoblast-like cells (re: sub-cluster 0, Suppl Fig 9, which as referred to as the “putative human anterior hypoblast” – page 9 paragraph 1, page 11 paragraph 2). The presumption that that the regionalized cells act as a signalling source is based on the expression profile of signal-response factors in the epiblast.

We thank the Reviewer for this comment and recognise this limitation. Thus, we refer to the anterior hypoblast as a putative, not a definitive, signalling centre throughout the text. Our study provides initial insights into the possibility of these hypoblast cells to act as a signalling centre by the expression of evolutionary conserved antagonists (Figure 3 c, d and i) and the correlation with patterning in the adject epiblast (Fig. 3m, Supplementary Fig. 10b). We recognise that future studies will be needed and introduced the following in the discussion:

<<Future studies will be needed to dissect the mechanism of action of this putative signalling centre for epiblast patterning and the establishment of the future AP axis.>>

As the asymmetrical distribution of the anterior hypoblast-like cells was observed in “a subset of embryos” can be gleaned from the dataset shown in Supp Fig 7. The pooled data (Fig. 3h) of a selected set of embryos pre-judged to have displayed such asymmetry did not add more to the inference of biased distribution of these hypoblast cells in some embryos.

We would like to clarify this point. In Figure 3h we have not selected a subset of embryos but included all the samples, regardless of their significance. We think this provides an unbiased visualisation of the distribution of hypoblasts cells among all the embryos analysed. We apologise for the confusion and we have clarified this in the figure legend.

Regarding the distribution of LEFTY-positive cells, which subset of embryos (only 2 among those in Suppl Fig 10?) showed the biased distribution? Were these embryos also displayed biased distribution of “CERL1-secreting” cells, in view of the single-cell result of co-expression of CERL1 and LEFTY1/2?

We recognise that only two embryos show a statistically significant bias distribution as shown in Suppl Figure 10a. It is important to note, however, that other embryos also display a clear biased trend (e.g., MZG414 and MZG 454), but this does not reach statistical significance. Therefore, similarly to CER1 analysis, we provided quantification for each individual embryo (Suppl Figure 10a) and a summary distribution including all the embryos (Figure 3I). It is important to note the limitation in defining significance by using one sample T-test with 90 degree as reference and the relatively small number of cells present per embryo and thus weak statistical power.

Unfortunately, we were not able to assess the co-expression of CER1 and LEFTY1/2 by immunofluorescence as the two primary antibodies are raised in the same species. Though we are unable to provide protein-level data due to these technical limitations, we have assessed the co-expression of CER1 and LEFTY1/2 transcriptionally as shown in Figure 3i.

Benchmarking the pluripotency state (Page 5-6): The characterization of cell state by similarities of transcriptome (“strongly resembled”) is equivocal. In essence, the epiblast cells of the 9 dpf and 11 dpf embryos are overall similar to H9 cells (and other conventional hESCs – Suppl Fig 5a), and they are also “significantly similar” (by correlation analysis) to H9 reset cells. The data did not support the inference of “indicate a progressive pluripotency transition” at 9 and 11 dpf. The epiblast cells in the cultured embryos are likely at an undefined (intermediate?) state (discussion: page 10 paragraph 1) that is shared in part with both H9 and H9 reset cells.

We thank the Reviewer for this comment. The progressive transition refers to pre-implantation (6-7 d.p.f) to post-implantation (11 d.p.f) development. Naïve (H9 Reset) and primed (H9) cells show significant similarities to pre- and post-implantation pluripotency respectively. We apologise for the confusion and have rephrased the text.

Overall, our data indicate a progressive pluripotency transition of the epiblast following implantation, in agreement with recent studies of human embryos²³, and expression profiles observed in mouse and monkey embryos^{7,8,47}.

Qualify the descriptor of “preferentially” expressed low level of other FGFs – would “differentially” be appropriate? (Page 6)

We have corrected this sentence.

What is the toxicity of “removing FGF signalling entirely” (Page 7)? Is dampening cell proliferation and thereby leading to reduced cell number a toxic effect? Supplementing exogenous FGF2 and FGF4 activity did not change cell number but it minimized the variation of epiblast cell number. Is a more consistent cell number expected of 9 and 11 dpf embryos *in vivo* (if such *in vivo* data are available)? Would the variation in epiblast and hypoblast cell number among the cultured embryos imply the culture has not provide optimal embryotrophic activity? How may this affect the identification of cell types, the relative abundance of specific cell type (if not all cells in the embryo have been sampled for analysis, Fig. 2d-f, Suppl Fig 6f-h), and the source and level of inter- cellular signalling activity?

We apologise for this mistake in the text and corrected as follows: <<we cannot rule out a potential toxic effect of the drug itself in addition to the effect of removing FGF-signalling entirely>> (page 7, first paragraph)

Such *in vivo* data at this stage of development do not exist, limiting the possibility to have a reference number for comparison. We acknowledge the limitation of the culture conditions as stated in the methods and that development *in vivo* would always be better but not possible for 2 week old human embryo. In our analyses we have only included embryos that contain both embryonic and extra-embryonic lineages, excluding suboptimally developed embryos as shown in Suppl Table 1.

The relative cell type abundance cannot be inferred by the bioinformatics analysis itself as a proportion of cells are lost during the sequencing pipeline (from encapsulation to final cell exclusion). The final cell number therefore does not necessarily reflect that original ratios present in the embryos. Therefore, we referred to relative abundance only on whole mount embryos analysed by immunofluorescence and reported these values as ratios between CER1+ cells and the number of hypoblast cells for normalisation.

Page 10-11: The results of the experiments of chemical inhibition and signalling factor supplementation did not warrant the inference of FGF signals are “emanating from the epiblast lineage” to regulate the development, in the context of proliferation or “fate specification” of three blastocyst cell lineages.

We have corrected as below:

<<However, here we found that FGF signals are necessary for growth and proliferation of the epiblast, hypoblast and trophoblast during early post-implantation development.>>

pSMAD1.5 results were shown in Fig. 3m, not 3l (page 9 paragraph 2)

We apologize for the error and have corrected it.

Suppl Fig 5 c and d were missing (page 5 second last line).

We apologize for the error and have corrected it.